# INTERPRETABLE INTRINSIC CUES FOR EFFICIENT REINFORCEMENT LEARNING WITH LARGE LANGUAGE MODELS

## ABSTRACT

Reinforcement learning with verifiable rewards (RLVR) has improved the reasoning ability of large language models, yet training remains costly because many rollouts contribute little to optimization relative to their heavy computational demands. This study investigates how simply leveraging interpretable and intrinsic data properties, which come at almost no additional computational cost during training, can markedly improve data efficiency for RLVR. We propose PREPO, an RLVR model with two complementary components. First, we use prompt perplexity as a proxy for model adaptability in learning, and adopt a schedule to guide the model from well-understood prompts to progressively challenging ones. Second, we amplify the diversity among rollouts by differentiating their relative entropy and prioritizing sequences with greater exploratory behavior. Together, these mechanisms reduce rollout demand while preserving competitive performance. On Qwen and Llama models, PREPO achieves effective results on mathematical reasoning benchmarks with up to $3\times$ fewer rollouts than baselines. Beyond empirical gains, we provide theoretical and in-depth analyses that explain how our method improves the data efficiency of RLVR.

## 1 INTRODUCTION

Reinforcement learning (RL) has become central in improving the reasoning capabilities of large language models (LLMs) by optimizing self-generated rollouts (Guo et al., 2025; Team et al., 2025; Chen et al., 2025a). Recent advances in reinforcement learning with verifiable reward (RLVR) demonstrate that it is a simple yet effective method for scaling reasoning performance (Shao et al., 2024; Yu et al., 2025). However, applying RLVR to models that generate long reasoning traces will incur substantial computational overhead in the rollout stage, significantly hampering RL training efficiency and becoming the primary training bottleneck (Zhong et al., 2024).

The exploration of effective strategies for leveraging data in RL training remains relatively under-developed. Prior study (Zhang et al., 2025; Albalak et al., 2025) has suggested using the pass rate as an indicator of data difficulty to strengthen the training signal. Nonetheless, this approach often requires multiple rounds of sampling to attain a sufficiently stable measurement. In fact, it converts online rollouts to an offline context, which ultimately does not contribute to reducing the overall computational burden. Other metrics, such as human-defined criteria (Chen et al., 2025b; Parashar et al., 2025), *e.g.*, specific domains or topics, have the disadvantage of being influenced by the tagging process, as well as the biases from human perceptions and experiences. In addition, alternative approaches often depend on auxiliary trained models or embedding techniques to reflect data semantics. We assert that these techniques not only pose significant demands on computational and memory cost, but also present mismatch issues and limited applicability for diverse types of policy models and training tasks. Furthermore, they overlook the inherent dynamism present in the RL training process, greatly lagging behind the pace of training updates. As a remedy, other methods incorporate historical information from training to reflect the dynamic nature of data. Nevertheless, it increases memory demands on the RL framework and includes extraneous noise during the training.

A natural way to improve training efficiency is through data selection, i.e., pruning uninformative prompts or rollouts while preserving those that drive learning. There are emerging approaches based

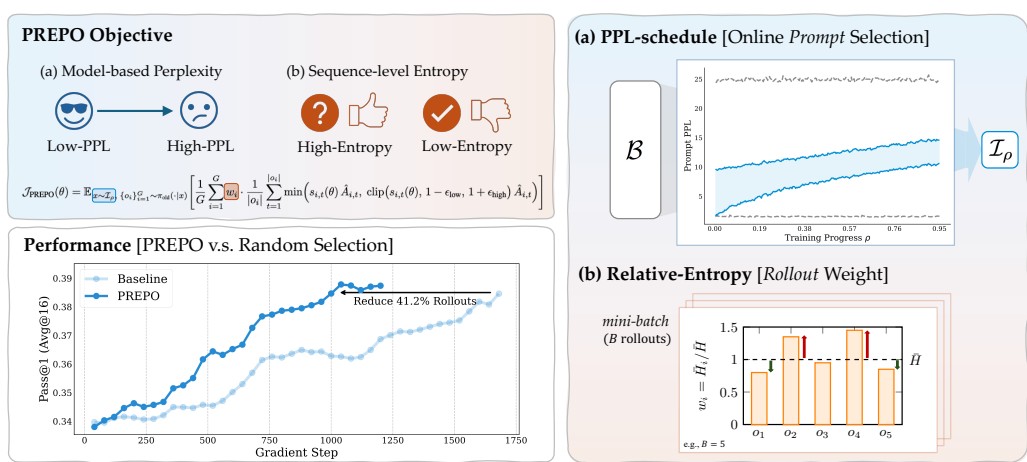

Figure 1: **Overview of PREPO.** The PREPO objective integrates perplexity-based schedule learning and sequence-level entropy weighting into a unified optimization scheme. On Qwen2.5-Math-7B, PREPO achieves higher performance while requiring only 41.2% of the rollouts used by random selection, showing improved efficiency. Specifically, PREPO has two complementary components: (a) PPL-schedule, which actively selects prompts according to model-based perplexity, starting with lower-PPL prompts *("adapted" to the model)* and progressively introducing higher-PPL ones *("obscured" to the model)* as training progresses. (b) Relative-Entropy Weighting, which adjusts rollout contributions by comparing each sequence's entropy against the batch average, amplifying the high-entropy rollouts *("novel attempts")* and downweighting the low-entropy *("regular response")*.

on parameterized modeling (Qu et al., 2025), replay buffers (Liu et al., 2025), or selective rollout execution (Zheng et al., 2025). Instead of aiding by inductive biases from both humans and external models, we address this long-standing research question from a new perspective:

*Can the intrinsic data properties deriving from the training process improve the efficiency of RLVR?*

In this study, we propose a simple method with almost negligible computation cost, Perplexity-schedule with Relative-Entropy Policy Optimization (PREPO), a method that combines a perplexity-based schedule with sequence-level entropy weighting to realize *intrinsic exploration*. Specifically, PREPO traces perplexity *before* rollout generation to prune the prompts, and applies entropy weighting *after* rollout generation to emphasize uncertain responses. It is worth noting that metrics collected during standard RL training process can be reused to compute our two components, ensuring the computational efficiency of our method. Moreover, our method is coherently integrated with the policy model and training process, offering a favorable trade-off between suitability and flexibility. Beyond that, our approach uncovers the intrinsic nature of data during RL training, providing fine-grained interpretability of training dynamics. Across Qwen and Llama models, PREPO surpasses existing data-pruning baselines and remains competitive with the baseline, while reducing rollout usage by more than 40% (see Fig. 1 for Qwen2.5-Math-7B). These results show that RLVR can be made substantially more efficient by leveraging the intrinsic properties of prompt and rollout data.

## 2 RELATED WORK

**Data efficiency in RLVR.** A growing body of work has explored data efficiency for RLVR, with particular attention to online data selection. Unlike offline methods that require pre-training or costly rollouts to estimate sample quality (Qu et al., 2025; Zhang et al., 2025; Chen et al., 2025b; Kamalloo et al., 2025), online approaches aim to reduce overhead by filtering or prioritizing samples dynamically during training. Online difficulty filtering (Bae et al., 2025) removes prompts that contribute little to reasoning improvement, while predictive prompt allocation (Qu et al., 2025) directs rollouts toward more promising inputs. Curriculum-based strategies further adapt training to the evolving competence of the model (Zhang et al., 2025; Chen et al., 2025b). Other online selection approaches prioritize samples using gradient-informed signals (Kamalloo et al., 2025; Chen et al., 2025c) or policy-advantage estimates (Wang and Guofeng, 2025). Parallel efforts focus on reducing

rollout redundancy during training. Down-sampling strategies (Li et al., 2025) and efficient replay buffer designs (Liu et al., 2025) lessen the burden of repeatedly training on uninformative samples. Collectively, these online methods emphasize the importance of allocating computational resources to samples that drive reasoning progress, a goal which aligns with the direction of our work.

**Entropy Mechanism in RLVR.** Entropy has long been studied in reinforcement learning through entropy-regularized objectives that promote exploration in control settings. Recent studies extend this idea to RLVR for reasoning LLMs. Cui et al. (2025) identify rapid entropy collapse as a major failure mode and propose covariance-based updates to slow its decay. Wang et al. (2025) show that high-entropy "forking tokens," though rare, account for most reasoning gains, highlighting entropy as a token-level signal of informativeness. Cheng et al. (2025) incorporates a clipped, gradient-detached entropy term into the advantage function, encouraging more exploratory responses but introducing additional hyperparameters. Building on these insights, we propose a parameter-free approach to reinforce entropy-driven exploration in RLVR. A related line of work explores weighting in policy-gradient updates through truncated importance sampling, such as CISPO (Chen et al., 2025a). These methods address a different dimension of RLVR optimization, and can be viewed as complementary to our focus on online data selection and entropy-based weighting.

## 3 PRELIMINARY ANALYSIS

### 3.1 LOW-PPL PROMPTS TEND TO YIELD HIGHER PASS RATE

We begin by examining the relationship between prompt perplexity (PPL) and task difficulty using the DAPO-Math-17K dataset (Yu et al., 2025). For both Qwen and Llama models, Figure 2 shows a clear negative correlation between PPL and passrate@16, where passrate@16 measures the fraction of prompts solved by at least one of 16 generations. Lower-PPL prompts generally yield higher success rates. Table 1 correlation is statistically significant across models, suggesting that PPL can serve as a lightweight signal to identify more informative prompts for training.

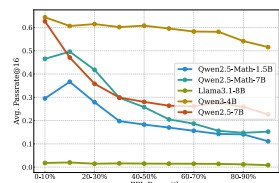

Figure 2: Prompt PPL versus average passrate@16.

Table 1: Correlation between prompt PPL and passrate@16. (*** $p < 0.001$, ** $p < 0.05$)

|  | Qwen2.5-7B | Qwen2.5-7B | Qwen2.5-1.5B | Qwen3-4B | LLama3.1-8B |
| --- | --- | --- | --- | --- | --- |
| Spearman | $-0.233$*** | $-0.183$** | $-0.186$*** | $-0.169$*** | $-0.199$*** |

### 3.2 TRAINING DYNAMICS OF LOW-PPL AND HIGH-PPL PROMPTS

To better understand the role of PPL during training, we compare prompts from the lowest 20% (LOW-PPL) and highest 20% (HIGH-PPL) of the distribution (see Appendix E for examples of the two groups). Figure 3 illustrates their training dynamics on Qwen2.5-Math-7B. The two groups exhibit complementary behavior: LOW-PPL prompts drive rapid improvements in reward and validation accuracy during early training, though at the cost of faster entropy collapse, whereas HIGH-PPL prompts preserve entropy and lower zero-advantage ratio, yielding stronger performance in later stages.

These trends are consistent across other Qwen models (Appendix C). Specifically, (a) HIGH-PPL prompts are associated with higher entropy, (b) LOW-PPL prompts yield higher rewards and validation accuracy early on, and (c) HIGH-PPL prompts maintain exploration and ultimately close the performance gap. For Llama3.1-8B (Figure 11), a similar pattern appears, though overall performance remains limited as the validation dataset is too challenging for Llama models.

### 3.3 COMPARISON WITH RANDOM SAMPLING

To test whether PPL-based grouping offers value beyond chance, we also compare with a random 20% subset. As shown in Figure 4, the random group consistently falls between the LOW-PPL and

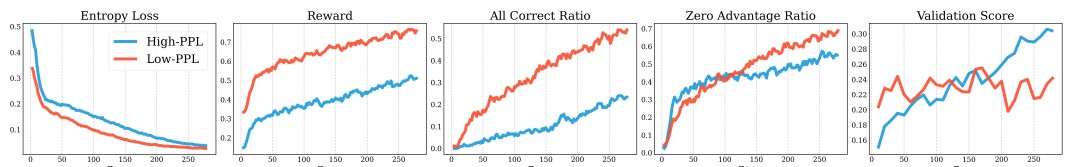

Figure 3: Training dynamics of LOW-PPL vs. HIGH-PPL prompts on Qwen2.5-Math-7B. *(a)* HIGH-PPL *prompts have higher entropy. (b)* LOW-PPL *prompts have more reward gains. (c)* LOW-PPL *prompts reach higher all-correct ratios faster. (d)* LOW-PPL *prompts show higher zero-advantage ratios in the later stage. (e)* HIGH-PPL *prompts eventually outperform* LOW-PPL *prompts.*

HIGH-PPL groups. This indicates that PPL is a non-trivial, policy-intrinsic signal that separates easy prompts with high-pass rates from more challenging ones, providing a useful basis for online batch selection design.

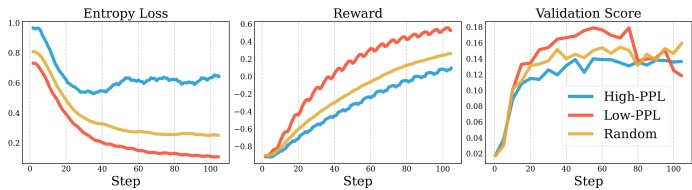

Figure 4: Comparison among LOW-PPL, HIGH-PPL, and Random Subsets. *Random lies between the two, showing that PPL-based grouping provides a meaningful pruning signal.*

## 4    PREPO: PPL-SCHEDULE RELATIVE-ENTROPY POLICY OPTIMIZATION

Building on the preliminary results, PREPO integrates a perplexity-driven schedule with entropy-based rollout weighting, forming a unified framework to improve efficiency in RLVR.

### 4.1    GENERAL ONLINE BATCH SELECTION

Let $\mathcal{B} = \{x_i\}_{i=1}^{N}$ denote the candidate batch at a training step. The goal of online batch selection is to design a mapping

$$\Phi : [0,1] \rightarrow 2^{\mathcal{B}}, \quad \rho \mapsto \mathcal{I}_\rho, \tag{1}$$

where $\rho \in [0,1]$ denotes the normalized training progress, and $\mathcal{I}_\rho \subseteq \mathcal{B}$. The mapping $\Phi$ is required to (i) explicitly depend on $\rho$, so that the distribution of selected samples evolves with training; (ii) the sub-batch size is fixed during training, i.e., $\forall \rho, |\mathcal{I}_\rho| = K$.

### 4.2    PPL-SCHEDULE ONLINE BATCH SELECTION

For a prompt $x_i = (x_{i,1}, \dots)$, we measure its perplexity under the policy $\pi_\rho$ at training progress $\rho$ as

$$P_i(\rho) = \exp\left(-\frac{1}{|x_i|} \sum_{t=1}^{|x_i|} \log \pi_\rho(x_{i,t} \mid x_{i,<t})\right), \tag{2}$$

where $\pi_\rho$ is the model distribution at progress $\rho$. As $\pi_\rho$ is parameterized by $\theta$ and evolves throughout training, $P_i(\rho)$ provides a model-based measure for active data selection. We then define the *PPL-schedule* sub-batch as

$$\mathcal{I}_\rho = \{\, \sigma(j) : l(\rho) \leq j \leq l(\rho) + K - 1 \,\}, \tag{3}$$

where $\sigma$ is the permutation that sorts $\mathcal{B}$ by ascending $P_i(\rho)$. The starting index $l(\rho)$ is given by a linear schedule

$$l(\rho) = \lfloor \rho \cdot (N - K) \rfloor, \tag{4}$$

so that $\mathcal{I}_\rho$ shifts smoothly from LOW-PPL to HIGH-PPL prompts. While linear scheduling is the simplest case, a nonlinear[1] one (e.g., quadratic or exponential) can also be used. In general, the PPL-schedule serves as an online data selection procedure that moves from more in-domain prompts to less in-domain prompts as training progresses.

## 4.3 RELATIVE ENTROPY WEIGHTING

As shown in Section 3.2, we empirically find that training on LOW-PPL prompts accelerates reward improvement but also leads to a rapid collapse of entropy, thereby reducing exploration. To mitigate this effect during the PPL schedule, we introduce a sequence-level relative-entropy weighting scheme that adaptively emphasizes uncertain rollouts.

The token-level entropy of a rollout is defined as $H_t = -\sum_{v \in \mathcal{V}} \pi_\theta(v \mid o_{<t}, x) \log \pi_\theta(v \mid o_{<t}, x)$, where $\mathcal{V}$ is the vocabulary. For rollout $i$, the sequence-level entropy is the average across its tokens

$$\bar{H}_i = \bar{H}(o_i \mid x) = \frac{1}{|o_i|} \sum_{t=1}^{|o_i|} H_t. \tag{5}$$

The batch-average entropy over $B$ rollouts is

$$\bar{H} = \frac{1}{B} \sum_{k=1}^{B} \bar{H}_k. \tag{6}$$

The relative weight assigned to rollout $i$ is then given by

$$w_i = \frac{\bar{H}_i}{\bar{H}}. \tag{7}$$

This formulation is scale-invariant, as a rollout's contribution depends only on its entropy relative to the batch mean. Intuitively, this design enables the model to *seek uncertainty within certainty* during the PPL schedule. While LOW-PPL prompts early in training often yield confident (low-entropy) responses, relative weighting amplifies the impact of less confident (higher-entropy) rollouts, thereby preserving exploration throughout training.

## 4.4 OBJECTIVE FUNCTION

The PREPO objective integrates PPL-schedule filtering with relative-entropy weighting as below.

$$\mathcal{J}_{\text{PREPO}}(\theta) = \mathbb{E}_{x \sim \mathcal{I}_\rho, \{o_i\}_{i=1}^{G} \sim \pi_{\text{old}}(\cdot|x)} \left[ \frac{1}{G} \sum_{i=1}^{G} w_i \cdot \frac{1}{|o_i|} \sum_{t=1}^{|o_i|} \min\left( s_{i,t}(\theta) \, \hat{A}_{i,t}, \, \text{clip}\left(s_{i,t}(\theta), 1 - \epsilon_{\text{low}}, 1 + \epsilon_{\text{high}}\right) \hat{A}_{i,t} \right) \right] \tag{8}$$

where $\mathcal{I}_\rho$ is the PPL-schedule-filtered batch of prompts at training progress $\rho$, $w_i$ encodes the relative entropy of rollout $i$ at the current micro-batch, $s_{i,t}(\theta)$ is the token-level importance ratio, $s_{i,t}(\theta) = \frac{\pi_\theta(o_{i,t}|x, o_{i,<t})}{\pi_{\text{old}}(o_{i,t}|x, o_{i,<t})}$, and $\hat{A}_{i,t}$ is the group-based advantage estimate, $\hat{A}_{i,t} = \frac{r_i - \text{mean}(\{r_j\}_{j=1}^{G})}{\text{std}(\{r_j\}_{j=1}^{G})}$. We set the clipping thresholds with $\epsilon_{\text{low}} = 0.2$ by default and a larger $\epsilon_{\text{high}} = 0.28$ for the upper bound.

## 4.5 PSEUDO-CODE OF PREPO ALGORITHM

Algorithm 1 summarizes the PREPO training loop. At each step, a candidate batch of prompts is sampled and filtered by the perplexity-based schedule to form the training subset. Rollouts from these prompts are then grouped into mini-batches, where relative-entropy weights are computed by normalizing each sequence entropy against the batch average. These weights amplify the contribution of high-entropy sequences when scaling the clipped objective. In this way, PREPO integrates schedule filtering with entropy-based weighting to implement the proposed objective.

---

[1]We define a general family of nonlinear schedules as $l(\rho) = \lfloor \rho^\alpha \cdot (N - K) \rfloor$, where $\alpha \in \mathbb{R}^+$. For example, choosing $\alpha = \frac{1}{2}g$ yields a square-root schedule that transitions more quickly toward higher-PPL prompts.

**Algorithm 1:** PPL-schedule Relative-Entropy Policy Optimization (PREPO)

---

**Input** : Dataset $\mathcal{D}$, actor $\pi_\theta$, candidate prompt size $N$, selected prompts $K$, rollouts per prompt $G$, mini-batch size $B$, total steps $T$, clipping $\epsilon_{\text{low}}, \epsilon_{\text{high}}$.

**Output** : Updated parameters $\theta$

**for** $t = 1, \ldots, T$ **do**

$\quad \rho \leftarrow \frac{t-1}{T-1}$;

$\quad B_t \leftarrow \text{Sample}(\mathcal{D}, N)$;

$\quad P_i \leftarrow \text{PPL}(x_i; \pi_\theta), \ x_i \in B_t$ ;$\qquad\qquad\qquad$ `// Prompt PPL Eq. (2)`

$\quad I_\rho \leftarrow \{\sigma(j) : l(\rho) \leq j < l(\rho) + K\}$ ;$\qquad\qquad$ `// PPL-schedule Eq. (3)`

$\quad \mathcal{U} \leftarrow \emptyset$;

$\quad$**for** $x \in I_\rho$ **do**

$\quad\quad$**for** $i = 1, \ldots, G$ **do**

$\quad\quad\quad\lfloor\ o \sim \pi_{\text{old}}(\cdot|x); r \leftarrow \text{Reward}(x, o)$ ;

$\quad\quad \hat{A} \leftarrow \frac{r-\mu}{\sigma}, \ \mu, \sigma$ over $G$ rollouts; Append $(x, o, \hat{A})$ to $\mathcal{U}$

$\quad M \leftarrow |\mathcal{U}|/B$; Partition $\mathcal{U}$ into $\{\mathcal{B}_j\}_{j=1}^M$ with $|\mathcal{B}_j| = B$ ;$\qquad$ `// Mini-batch split`

$\quad \mathcal{L} \leftarrow 0$;

$\quad$**for** $j = 1, \ldots, M$ **do**

$\quad\quad$**foreach** $(x, o, \hat{A}) \in \mathcal{B}_j$ **do**

$\quad\quad\quad\lfloor\ \bar{H} \leftarrow \frac{1}{|o|}\sum_t\left(-\sum_{v \in \mathcal{V}} \pi_\theta(v|o_{<t}, x) \log \pi_\theta(v|o_{<t}, x)\right)$ ; `// Sequence entropy Eq. (5)`

$\quad\quad \bar{H}^{(j)} \leftarrow \frac{1}{B}\sum_{(x,o,\bar{H},\hat{A}) \in \mathcal{B}_j} \bar{H}$ ;$\qquad\qquad\qquad$ `// Batch entropy Eq. (6)`

$\quad\quad$**foreach** $(x, o, \bar{H}, \hat{A}) \in \mathcal{B}_j$ **do**

$\quad\quad\quad\lfloor\ w \leftarrow \frac{\bar{H}}{\bar{H}^{(j)}}$ ;$\qquad\qquad\qquad\qquad$ `// Relative entropy weight Eq. (7)`

$\quad\quad J^{(j)}(\theta) \leftarrow \frac{1}{|\mathcal{B}_j|}\sum_{(x,o,\hat{A},w) \in \mathcal{B}_j} w \cdot \frac{1}{|o|}\sum_t \min\left(s_t\hat{A}, \text{clip}(s_t, 1-\epsilon_{\text{low}}, 1+\epsilon_{\text{high}})\hat{A}\right)$

$\quad\quad \mathcal{L} \leftarrow \mathcal{L} + \frac{1}{M}J^{(j)}(\theta)$

$\quad \theta \leftarrow \theta + \eta\nabla_\theta\mathcal{L}$

**return** $\theta$

---

## 5 EXPERIMENTS

### 5.1 SETUPS

We benchmark PREPO against three baselines: Random selection, Dynamic Sampling (DS) (Yu et al., 2025), and GRESO (Zheng et al., 2025) (detailed in Appendix F). Our evaluation covers multiple models: Qwen2.5-7B (Team, 2024), Qwen2.5-Math-1.5B and Qwen2.5-Math-7B (Yang et al., 2024), Qwen3-4B (non-thinking) (Yang et al., 2025), and Llama3.1-8B (Dubey et al., 2024). For training data, we use DAPO-Math-17K (Yu et al., 2025) and MATH500 (Lightman et al., 2023a).

**Training and Evaluation.** All models are trained using the `verl` (Sheng et al., 2025), with `vLLM` (Kwon et al., 2023) employed for rollout generation to ensure efficient inference. For the Qwen models, we evaluate them on four benchmarks, including AIME25 (Art of Problem Solving, 2025), AIME24 (Art of Problem Solving, 2024), MATH500 (Lightman et al., 2023b), and OlympiadBench (He et al., 2024), which cover a diverse range of mathematical reasoning challenges. The Llama model is evaluated on MATH500 (Lightman et al., 2023b) and GSM8K (Cobbe et al., 2021). We evaluate all models using *pass@1 (avg16)*, i.e., the accuracy of the top-1 response averaged over 16 generations, with temperature 1. We evaluate models every 50 training steps and report the best average performance on all benchmarks.

**Experiment Configuration.** For the Qwen2.5-Math models, we use a maximum context length of 4096 tokens, matching their supported limit. For Qwen3-4B and Llama3.1-8B, we set the context length to 32,768 tokens. Rollouts are generated with temperature as 1 using `vLLM`, producing 8 responses per prompt. For PREPO, Random, and GRESO, we adopt an online selection ratio of $K/N = 20\%$ at each training step, where the candidate batch size ($N$) is 1280, with the actual batch size ($K$) fixed at 256. For GRESO, we set the targeted zero-variance percentage as $50\%$. For DS, the

candidate batch size is 384. The mini-batch size ($B$) is 64 for all experiments. The actor model is optimized with AdamW using a constant learning rate of $1 \times 10^{-6}$, momentum parameters $\beta_1 = 0.9$ and $\beta_2 = 0.999$, and a weight decay of 0.01. Following Yu et al. (2025), we omit the KL-divergence regularization term. Training is applied only to the actor parameters and parallelized with Fully Sharded Data Parallel. All experiments are conducted on 32 GPUs.

Table 2: Performance comparison (%) on Qwen models. *For each model, the top row reports the base model's performance. Best results are highlighted in bold or underlined. K = thousand, M = million.*

| Method | AIME25 | AIME24 | MATH | Olympiad | Avg ↑ | # Rollouts ↓ |
|---|---|---|---|---|---|---|
| *Qwen2.5-7B* | 1.25 | 4.17 | 72.26 | 33.09 | 27.69 | – |
| + DS | 7.92 | 17.55 | 75.50 | 38.91 | 34.97 | 1040K |
| + Random | 6.98 | 16.41 | 75.70 | 38.47 | 34.39 | 716K |
| + GRESO | 9.22 | 10.83 | 76.65 | 42.07 | 34.59 | 680K |
| + PREPO (Ours) | 10.21 | 16.09 | 76.30 | 39.85 | **35.61** | **304K** |
| *Qwen2.5-Math-1.5B* | 3.54 | 10.21 | 55.76 | 27.41 | 24.23 | – |
| + DS | 10.83 | 25.83 | 76.40 | 23.33 | 34.10 | 3.6M |
| + Random | 20.00 | 16.67 | 76.25 | 30.50 | 35.86 | 3.0M |
| + GRESO | 15.38 | 20.00 | 76.65 | 24.17 | 34.16 | 2.5M |
| + PREPO (Ours) | 20.00 | 16.67 | 76.25 | 32.00 | **36.23** | **1.1M** |
| *Qwen2.5-Math-7B* | 9.17 | 20.80 | 72.26 | 39.56 | 35.45 | – |
| + DS | 13.33 | 33.33 | 81.35 | 30.17 | 39.55 | 1664K |
| + Random | 10.00 | 26.67 | 77.80 | 43.26 | 39.45 | 905K |
| + GRESO | 18.33 | 25.83 | 77.80 | 26.83 | 37.46 | 654K |
| + PREPO (Ours) | 12.81 | 26.15 | 77.85 | 41.58 | **39.59** | **540K** |
| *Qwen3-4B* | 30.00 | 53.33 | 94.10 | 52.67 | 57.53 | – |
| + DS | 63.33 | 66.67 | 95.10 | 58.00 | 70.78 | 688K |
| + Random | 60.00 | 70.00 | 96.00 | 59.33 | 71.33 | 553K |
| + GRESO | 56.67 | 69.17 | 96.40 | 57.33 | 69.89 | 472K |
| + PREPO (Ours) | 66.67 | 80.00 | 96.60 | 60.67 | **75.99** | **348K** |

## 5.2 RESULTS

**PREPO achieves consistent two- to three-fold rollout reduction.** As shown in Table 7, PREPO substantially lowers rollout usage relative to all baselines while maintaining or surpassing accuracy. On Qwen2.5-7B, PREPO reduces rollouts from over 1M (DS: 1,040K; Random: 716K; GRESO: 680K) to 304K. On Qwen2.5-Math-1.5B, it cuts the budget from several million (DS: 3.6M; Random: 3.0M; GRESO: 2.5M) down to 1.1M. Similarly, PREPO reduces rollouts to 540K on Qwen2.5-Math-7B (vs. 1.66M for DS, 905K for Random, 654K for GRESO) and to 348K on Qwen3-4B (vs. 688K, 553K, and 472K, respectively).

**Dynamic sampling and random selection are inefficient.** Although DS sometimes yields accuracy gains, it consistently demands the largest rollout budget. For instance, on Qwen2.5-Math-1.5B, DS requires 3.6M rollouts to reach an average score of 34.10, whereas PREPO attains a higher 36.23 with only 1.1M rollouts. This confirms that DS wastes computation by discarding uninformative prompts only after rollouts are generated. Random selection occasionally performs comparably to or even better than DS, yet its rollout cost remains high. On Qwen2.5-Math-7B, for example, random selection consumes 905K rollouts to achieve 39.45, while PREPO surpasses it with 39.59 using just 540K rollouts. Overall, PREPO delivers both higher accuracy and lower cost, while also producing a more diverse set of problems than online random selection (see Appendix H.6).

**GRESO improves efficiency but lags behind PREPO.** GRESO reduces rollout demand by pre-filtering uninformative prompts, making it more efficient than DS and Random. However, its accuracy often falls short of PREPO. For instance, on Qwen3-4B, GRESO achieves 69.89 accuracy with 472K rollouts, while PREPO reaches 75.99 with only 348K rollouts. This suggests that PREPO's intrinsic exploration signals provide a more reliable and lightweight alternative to heuristic rollout filtering.

**PREPO generalizes across model architectures.** On Llama3.1-8B (Table 3), PREPO once again achieves the strongest results, reaching an average of 36.55 with just 115K rollouts. In contrast, DS requires nearly five times more rollouts with substantially lower accuracy. This confirms that PREPO generalizes effectively across model families and scales, providing a consistent advantage in both performance and efficiency.

Table 3: Performance comparison (%) on Llama. *For each model, the top row reports the base model's performance. Best results are highlighted in bold or underlined. K = thousand.*

| Method | GSM8K | MATH | Avg ↑ | # Rollouts ↓ |
|---|---|---|---|---|
| *Llama3.1-8B* | 9.53 | 6.05 | 7.79 | – |
| + DS | 39.50 | 17.00 | 28.25 | 553K |
| + Random | 46.63 | 14.60 | 30.61 | 266K |
| + GRESO | 41.77 | 16.80 | 29.29 | 273K |
| + PREPO (Ours) | 51.10 | 21.81 | **36.55** | **115K** |

**PREPO could be more effective when perplexity distributions are concentrated.** We found that PREPO achieves large improvement when perplexity values are relatively compact across Qwen3-4B, and LLaMA3.1-8B. As shown in Table 4, the normalized standard deviation of those models remains below one, indicating less dispersed distributions that make perplexity-based filtering more reliable. In these cases, PREPO can better exploit the PPL-schedule to reduce rollouts.

Table 4: Normalized standard deviation of prompt perplexity (std/mean) across models. *Values above one indicate more dispersed distributions.*

| Model | Qwen2.5-7B | Qwen3-4B | LLaMA3.1-8B | Qwen2.5-Math-7B | Qwen2.5-Math-1.5B |
|---|---|---|---|---|---|
| std/mean | 0.73 | 0.65 | 0.75 | 1.23 | 1.02 |

**Training Dynamics of PREPO versus the Random Baseline** Figures 5 and 14 present a comparison of training dynamics between PREPO and random selection. PREPO shows a higher entropy loss, indicating stronger exploratory behavior throughout training. It also sustains a higher gradient norm while avoiding instability, suggesting more active yet controlled parameter updates. In terms of learning efficiency, PREPO reduces the proportion of rollouts with zero advantage, thereby providing more informative gradients for optimization. Furthermore, the average prompt length under PREPO decreases steadily, implying an adaptive shift from longer to shorter prompts over time. A similar trend is observed in the average response length, where PREPO generates longer outputs than the baseline across most steps on average, reflecting a longer thinking behavior.

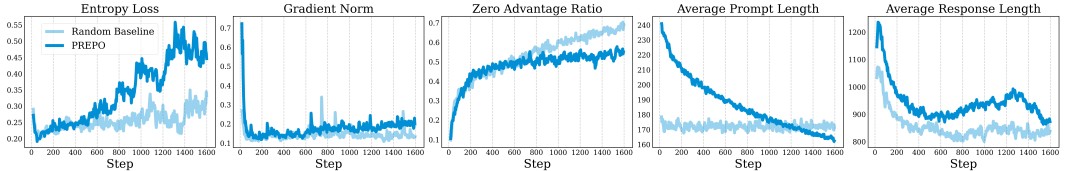

Figure 5: Full Comparison between PREPO and Random Selection on Qwen2.5-Math-7B.

## 5.3 ALTERNATIVE SELECTION RATIO

Figure 6 and Table 8 (see Appendix G) show the effect of varying the online batch selection ratio ($K/N$) with $N$ fixed for PREPO. The 20% ratio achieves the best average performance while also requiring the fewest rollouts. A 30% ratio is competitive on some benchmarks but less efficient, while 25% does not improve overall accuracy. Smaller ratios (15%, 10%, 5%) degrade performance and consume more rollouts. Overall, 20% strikes the best balance between accuracy and efficiency, and is adopted as the default in our experiments.

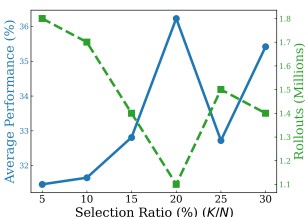

Figure 6: Different $K/N$

## 5.4 Ablation Study

In this section, we conduct the ablation analysis to isolate the contribution of each component in PREPO. In addition to the linear PPL-schedule, we evaluate four variants: (1) linear PPL-schedule, (2) random filtering with relative-entropy weighting, (3) linear PPL-schedule combined with an entropy-regularization loss, and (4) a nonlinear PPL-schedule. The nonlinear variant follows the general form $l(\rho) = \lfloor \rho^\alpha (N - K) \rfloor$ with $\alpha = 0.5$, which produces a schedule that shifts more quickly toward higher-PPL prompts.

As shown in Tables 5 and 6, the full PREPO method achieves the highest average performance across all model scales. Variants that remove or modify either component show consistent performance drops, indicating that both the perplexity-based schedule and the relative-entropy weighting contribute to PREPO's effectiveness. The nonlinear schedule performs competitively but remains slightly below the linear schedule, suggesting that the linear form is already a stable and effective design choice.

Table 5: Ablation study on Qwen. Performance comparison (%) between PREPO and (1) linear PPL-schedule, (2) random filtering with relative-entropy, (3) linear PPL-schedule with entropy loss, (4) non-linear PPL-schedule with relative-entropy. *Best results are highlighted in bold or underlined.*

| Model | Method | AIME25 | AIME24 | MATH | Olympiad Bench | Avg ↑ |
|---|---|---|---|---|---|---|
| Qwen2.5-Math-7B | PREPO | 12.81 | 26.15 | 77.80 | 41.58 | **39.59** |
|  | w/o relative entropy | 10.00 | 23.33 | 74.60 | 39.21 | 36.79 |
|  | w/o PPL-schedule | 11.87 | 25.73 | 76.48 | 34.43 | 37.13 |
|  | w/ entropy loss | 10.73 | 23.54 | 75.25 | 38.81 | 37.08 |
|  | w/ non-linear schedule | 12.71 | 26.15 | 76.40 | 40.12 | 38.85 |
| Qwen2.5-7B | PREPO | 10.20 | 16.09 | 76.30 | 39.85 | **35.61** |
|  | w/o relative entropy | 6.98 | 16.41 | 75.70 | 38.47 | 34.39 |
|  | w/o PPL-schedule | 8.89 | 16.04 | 79.41 | 22.54 | 31.72 |
|  | w/ entropy loss | 6.25 | 16.35 | 77.36 | 21.04 | 30.25 |
|  | w/ non-linear schedule | 9.58 | 16.25 | 76.31 | 39.60 | 35.44 |
| Qwen2.5-Math-1.5B | PREPO | 20.00 | 16.67 | 76.25 | 32.00 | **36.23** |
|  | w/o relative entropy | 10.21 | 15.68 | 72.10 | 30.50 | 32.12 |
|  | w/o PPL-schedule | 11.81 | 13.12 | 70.21 | 30.02 | 31.29 |
|  | w/ entropy loss | 6.46 | 16.56 | 73.77 | 30.99 | 31.95 |
|  | w/ non-linear schedule | 9.15 | 15.83 | 75.85 | 30.26 | 32.77 |
| Qwen3-4B | PREPO | 66.67 | 80.00 | 96.60 | 60.67 | 75.99 |
|  | w/o relative entropy | 64.77 | 72.70 | 90.54 | 59.06 | 71.77 |
|  | w/o PPL-schedule | 64.99 | 77.73 | 95.03 | 57.86 | 73.90 |
|  | w/ entropy loss | 61.10 | 75.17 | 88.39 | 60.54 | 71.28 |
|  | w/ non-linear schedule | 61.45 | 74.47 | 92.37 | 60.67 | 72.24 |

Table 6: Ablation study on Llama. Performance comparison (%) between PREPO and (1) linear PPL-schedule, (2) random filtering with relative-entropy, (3) linear PPL-schedule with entropy loss, (4) non-linear PPL-schedule with relative-entropy. *Best results are highlighted in bold or underlined.*

| Model | Method | GSM8K | MATH | Avg ↑ |
|---|---|---|---|---|
| Llama3.1-8B | PREPO | 51.10 | 21.81 | **36.55** |
|  | w/o relative entropy | 46.85 | 18.25 | 32.55 |
|  | w/o PPL-schedule | 34.55 | 16.29 | 25.42 |
|  | w/ entropy loss | 4.45 | 8.30 | 6.33 |
|  | w/ non-linear schedule | 46.85 | 20.56 | 33.71 |

## 5.5 Analysis of Relative-Entropy Weights

**Effective Batch Sizes** Relative-entropy weighting does not increase the effective batch size (i.e., $\frac{1}{B} \sum_i w_i$); it merely redistributes gradient contributions across sequences. Since the weights are

normalized by $\bar{H}$, we obtain

$$\frac{1}{B}\sum_{i=1}^{B} w_i \cdot |o_i| = \frac{1}{B\bar{H}}\sum_{i=1}^{B}|o_i|\bar{H}_i = \frac{1}{B\bar{H}}\sum_{i=1}^{B}\sum_{t=1}^{|o_i|}H_{i,t} = \frac{\sum_{i=1}^{B}|o_i|}{B}. \quad (9)$$

**Remark**: The token-weighted average weight equals the average sequence length. If all sequences have equal length, then $\frac{1}{B}\sum_i w_i = 1$. As shown in Figure 7, the effective batch size for PREPO on Qwen2.5-Math-1.5B stays close to $B$ throughout training. At the early steps, LOW-PPL prompts yield low-entropy responses, so higher-entropy rollouts receive more weight, pushing the average above $B$. As training advances and higher-PPL prompts are introduced, the overall entropy rises, and normalization shifts the average slightly below $B$.

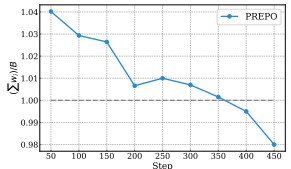

Figure 7: Trend of Effective Batch Size

**Sensitivity of Large Entropy**  Since $w_i = \bar{H}_i/\bar{H}$ is normalized by the batch mean, a rollout with $\bar{H}_j \gg \bar{H}$ has a large weight $w_j \gg 1$, while simultaneously reducing the weights of the others ($w_i! \ll !1$ for $i \neq j$). The derivative

$$\frac{\partial w_i}{\partial \bar{H}_j} = \begin{cases} \dfrac{1}{\bar{H}} - \dfrac{\bar{H}_j}{\bar{H}^2}\dfrac{|o_j|}{\sum_k |o_k|}, & i = j, \\[2ex] -\dfrac{\bar{H}_i}{\bar{H}^2}\dfrac{|o_j|}{\sum_k |o_k|}, & i \neq j, \end{cases}$$

clarifies this behavior: (1) for $i = j$, the two terms partially cancel, so although $w_j$ is already large, its growth rate is moderated as $\bar{H}_j$ increases further; (2) for $i \neq j$, the derivative is negative, confirming that a large $\bar{H}_j$ suppresses the weights of all other rollouts. Empirically, such outliers are rare: in PREPO on Qwen2.5-Math-1.5B at step 50, the weight distribution (Figure 8) shows most rollouts clustered near one, with negligible mass on extreme values. Further discussion is provided in Appendix H.

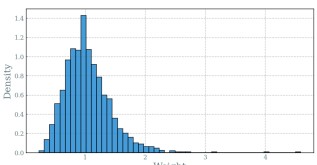

Figure 8: Frequency of Weights

## 6 CONCLUSION AND FUTURE WORK

This study investigated how intrinsic data properties can improve the efficiency of RLVR training. Prompt perplexity enables a natural schedule from easier to harder prompts, while sequence-level relative entropy amplifies exploratory rollouts. Integrated in PREPO, these components reduce rollout cost while maintaining or improving benchmark performance.

Beyond empirical gains, PREPO shows that RLVR can be guided by interpretable, policy-intrinsic cues rather than human heuristic or auxiliary models. This makes PREPO a practical and transparent recipe for compute-efficient RLVR, accessible even to researchers with limited resources. Future work may explore additional intrinsic signals (e.g., input token size) and combine data-driven exploration with system-level optimizations. Limitations of this study are discussed in Appendix A.

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

# Appendices

# A LIMITATIONS

This study has several limitations that should be acknowledged. (1) response lengths were restricted to 32K tokens, leaving the applicability of PREPO to models generating substantially longer outputs an open question; and (2) the evaluation was limited to mathematical reasoning tasks, while its effectiveness in other domains remains to be explored.

# B DISCLAIMER ON LLM USAGE

The use of LLMs is permitted as a general-purpose assistance tool. In this work, LLMs were employed solely for grammar correction and sentence rephrasing. They had no involvement in research ideation, experimental design, data analysis, or substantive writing. Their role was restricted to improving clarity and style; therefore, they are not considered contributors to the research.

# C MORE TRAINING DYNAMICS OF LOW-PPL AND HIGH-PPL PROMPTS

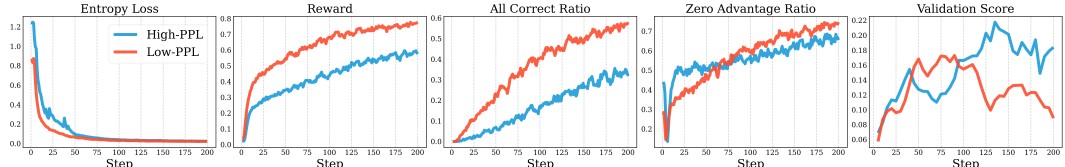

Figure 9: Training Dynamics of LOW-PPL vs. HIGH-PPL Prompts on Qwen2.5-7B.

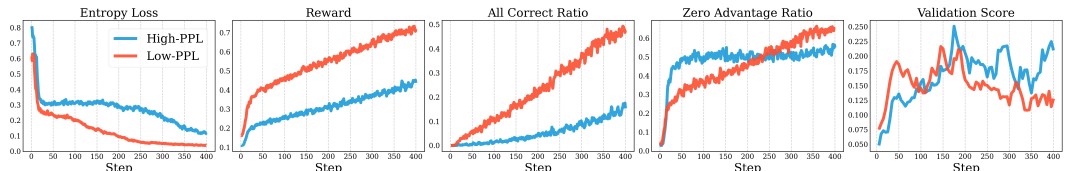

Figure 10: Training Dynamics of LOW-PPL vs. HIGH-PPL Prompts on Qwen2.5-Math-1.5B.

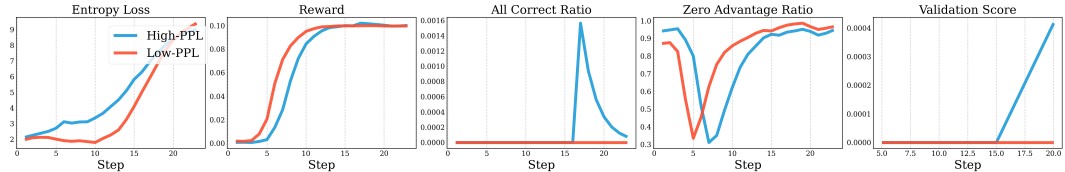

Figure 11: Training Dynamics of LOW-PPL vs. HIGH-PPL Prompts on Llama3.1-8B.

# D ENTROPY AS A RESPONSE CONFIDENCE IN ROLLOUTS

Standard RLVR methods rely solely on reward feedback and lack an explicit mechanism for assessing the confidence of generated responses. Token-level entropy serves as a common measure of confidence, as it quantifies the sharpness of the predictive distribution along a rollout.

Specifically, the token-level entropy of a rollout is defined as $H_t = -\sum_{v \in \mathcal{V}} \pi_\theta(v \mid o_{<t}, x) \log \pi_\theta(v \mid o_{<t}, x)$, where $\mathcal{V}$ is the vocabulary. The sequence-level entropy is then the average

$$\bar{H}(o \mid x) = \frac{1}{|o|} \sum_{t=1}^{|o|} H_t. \tag{10}$$

Rollouts with low entropy correspond to highly concentrated predictive distributions, reflecting strong model confidence in a single continuation. In contrast, rollouts with high entropy correspond to more diffuse distributions, where multiple continuations remain plausible.

Thus, entropy complements prompt filtering by providing an intrinsic measure of response confidence that can be directly incorporated into the training process.

# E  HIGH-PPL AND LOW-PPL PROMPTS

As illustrated in Figure 13, HIGH-PPL prompts exhibit a greater prevalence of non-English characters relative to LOW-PPL prompts. This pattern is consistently observed across both the Qwen2.5-series and Llama model families.

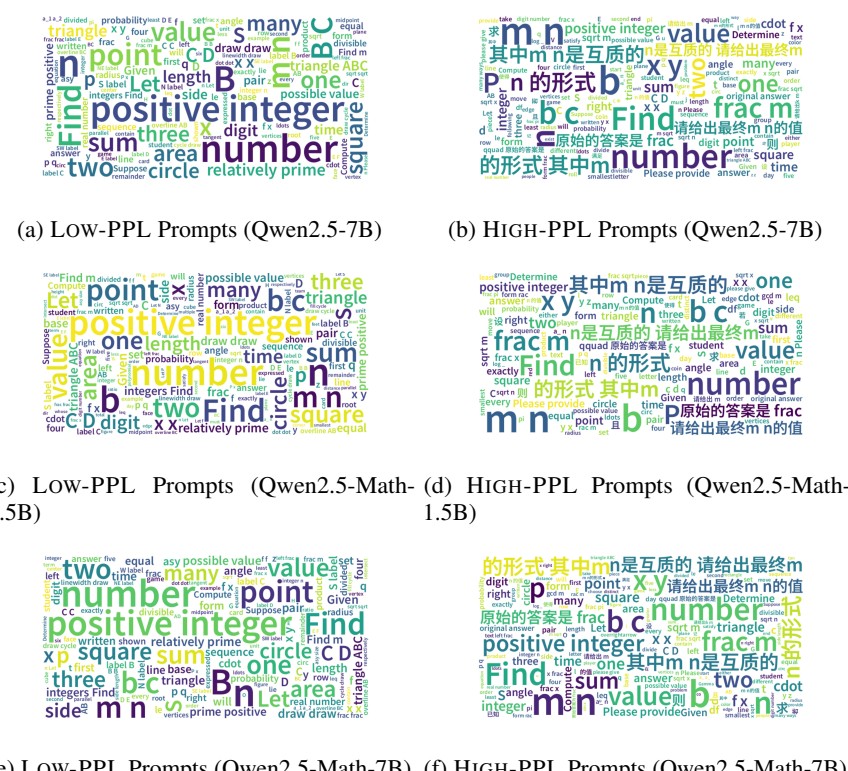

(a) LOW-PPL Prompts (Qwen2.5-7B)    (b) HIGH-PPL Prompts (Qwen2.5-7B)

(c) LOW-PPL Prompts (Qwen2.5-Math-1.5B)    (d) HIGH-PPL Prompts (Qwen2.5-Math-1.5B)

(e) LOW-PPL Prompts (Qwen2.5-Math-7B)    (f) HIGH-PPL Prompts (Qwen2.5-Math-7B)

Figure 12: Wordcloud of the Most Frequent Words in Low-/HIGH-PPL Prompts

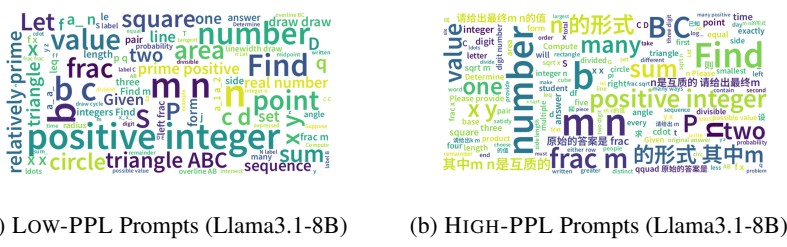

(a) LOW-PPL Prompts (Llama3.1-8B)    (b) HIGH-PPL Prompts (Llama3.1-8B)

Figure 13: Wordcloud of the Most Frequent Words in Low-/HIGH-PPL Prompts

**Example of LOW-PPL Problems**

- Cube $ABCDEFGH$, labeled as shown below, has edge length 1 and is cut by a plane passing through vertex $D$ and the midpoints $M$ and $N$ of $\overline{AB}$ and $\overline{CG}$ respectively. The plane divides the cube into two solids. Find the volume of the larger of the two solids. [asy] import cse5; unitsize(8mm); pathpen=black; pair A = (0,0), B = (3.8,0), C = (5.876,1.564), D = (2.076,1.564), E = (0,3.8), F = (3.8,3.8), G = (5.876,5.364), H = (2.076,5.364), M = (1.9,0), N = (5.876,3.465); pair[] dotted = A,B,C,D,E,F,G,H,M,N; D(A–B–C–G–H–E–A); D(E–F–B); D(F–G); pathpen=dashed; D(A–D–H); D(D–C); dot(dotted); label("$A$",A,SW); label("$B$",B,S); label("$C$",C,SE); label("$D$",D,NW); label("$E$",E,W); label("$F$",F,SE); label("$G$",G,NE); label("$H$",H,NW); label("$M$",M,S); label("$N$",N,NE); [/asy]The answer is in the form
x0cracmn, where gcd(m, n) = 1. Please provide the value of m + n.

- The number $a = \frac{p}{q}$, where $p$ and $q$ are relatively prime positive integers, has the property that the sum of all real numbers $x$ satisfying

$$\lfloor x \rfloor \cdot \{x\} = a \cdot x^2$$

is 420, where $\lfloor x \rfloor$ denotes the greatest integer less than or equal to $x$ and $\{x\} = x - \lfloor x \rfloor$ denotes the fractional part of $x$. What is $p + q$?

- Let $a, b$ be positive integers satisfying

$$\sqrt{\frac{ab}{2b^2 - a}} = \frac{a + 2b}{4b}$$

. Find $|10(a - 5)(b - 15)| + 8$.

- What is the greatest common divisor of $121^2 + 233^2 + 345^2$ and $120^2 + 232^2 + 346^2$?

**Example of HIGH-PPL Problems**

- At the Lexington High School, each student is given a unique five-character ID consisting of uppercase letters. Compute the number of possible IDs that contain the string "LMT".

- 设3 阶实对称矩阵 $A$ 的三个特征值分别为 $-1, -1, 2$，且 $(1, 1, -1)^T$ 是特征值2 所对应的特征向量. 记 $A$ 中所有元素的平方和为 $I$，则 $[I]$=__________.  (English: Let the eigenvalues of A real symmetric matrix of order 3 be $-1, -1, 2$, respectively, and $(1, 1, -1)^T$ be the eigenvector corresponding to eigenvalue 2. For the sum of squares of all elements in the A $I$, is $[I]$ = __________.)

- 求六个元素的置换群 $S_6$ 中6 阶元素的个数。 (English: Find the number of elements of order 6 in the permutation group $S_6$ of six elements.)

- Three bells begin to ring simultaneously. The intervals between strikes for these bells are, respectively, $\frac{4}{3}$ seconds, $\frac{5}{3}$ seconds, and 2 seconds. Impacts that coincide in time are perceived as one. How many beats will be heard in 1 minute? (Include first and last.)

## F  DESCRIPTION OF BASELINE METHODS

We compare PREPO against the following three baseline strategies.

- **Dynamic Sampling (DS).** Dynamic Sampling, as introduced in DAPO (Yu et al., 2025), dynamically filters out prompt groups whose generated responses all produce identical rewards (i.e. zero variance). In each training batch, DS resamples such uninformative prompt groups, thereby ensuring that the batch maintains a sufficient proportion of prompts that give meaningful gradient signals. However, it can still incur high rollout costs because many sampled prompts may remain uninformative until they are filtered.

- **Random.** The Random baseline uniformly selects prompts and associated rollouts without regard to historical feedback or variance in reward. All prompts are treated equally, so there is no mechanism

to avoid rollouts on uninformative or zero-variance prompts. This method serves as a lower bound in terms of data selection sophistication.

- **GRESO.** GRESO (GRPO with Efficient Selective Rollout) (Zheng et al., 2025) is a lightweight, online, pre-rollout filtering approach. It uses statistics of reward dynamics over previous epochs to predict which prompts are likely to be uninformative (e.g. zero variance among responses) and skips them before performing rollouts.

## G  ADDITIONAL EXPERIMENTAL RESULTS

Table 7: Details of performance comparison (%) on Qwen models. *Avg (avg@16/32/64) means the average of AIME25 (avg@16/32/64), AIME24 (avg16/32/64), MATH, and Olympiad. Best results are highlighted in bold or underlined.*

| Method | AIME25 (avg@16) | AIME25 (avg@32) | AIME25 (avg@64) | AIME24 (avg@16) | AIME24 (avg@32) | AIME24 (avg@64) | Avg ↑ (avg@16) | Avg ↑ (avg@32) | Avg ↑ (avg@64) |
|---|---|---|---|---|---|---|---|---|---|
| *Qwen2.5-7B* | 1.25 | 2.60 | 1.17 | 4.17 | 4.90 | 4.95 | 27.69 | 28.21 | 27.87 |
| + DS | 7.92 | 7.08 | 7.19 | 17.55 | 15.00 | 14.90 | 34.97 | 34.76 | 34.13 |
| + Random | 6.98 | 6.35 | 6.51 | 16.41 | 15.63 | 15.89 | 34.39 | 34.04 | 34.14 |
| + GRESO | 9.22 | 7.29 | 7.50 | 10.83 | 13.65 | 13.39 | 34.39 | 34.92 | 34.90 |
| + PREPO (Ours) | 10.21 | 10.62 | 10.00 | 16.09 | 15.52 | 16.35 | **35.61** | **35.72** | **35.63** |
| *Qwen2.5-Math-1.5B* | 3.54 | 3.75 | 4.48 | 10.21 | 8.33 | 8.96 | 24.23 | 23.81 | 24.15 |
| + DS | 10.83 | 13.76 | 14.54 | 25.83 | 21.72 | 23.04 | 34.10 | 33.80 | 34.33 |
| + Random | 20.00 | 17.45 | 18.81 | 16.67 | 15.42 | 16.50 | 35.86 | 34.91 | 35.52 |
| + GRESO | 15.38 | 11.40 | 15.39 | 20.00 | 19.20 | 17.97 | 34.16 | 32.86 | 33.56 |
| + PREPO (Ours) | 20.00 | 19.69 | 19.38 | 16.67 | 16.98 | 17.03 | **36.23** | **36.23** | **36.17** |
| *Qwen2.5-Math-7B* | 8.95 | 9.17 | 8.80 | 17.50 | 20.80 | 19.22 | 35.45 | 35.45 | 34.96 |
| + DS | 13.33 | 14.27 | 14.84 | 33.33 | 35.63 | 34.69 | 34.10 | 40.36 | 40.26 |
| + Random | 10.00 | 12.50 | 12.50 | 26.67 | 29.34 | 29.87 | 35.86 | 40.73 | 40.86 |
| + GRESO | 18.33 | 18.58 | 18.58 | 25.83 | 24.38 | 25.00 | 37.46 | 36.90 | 37.05 |
| + PREPO (Ours) | 12.81 | 14.41 | 15.93 | 26.15 | 29.17 | 29.25 | **39.59** | **40.75** | **41.15** |
| *Qwen3-4B* | 30.00 | 47.30 | 44.89 | 53.33 | 61.60 | 28.70 | 57.53 | 63.92 | 55.09 |
| + DS | 63.33 | 53.90 | 54.80 | 66.67 | 63.50 | 65.30 | 70.78 | 67.63 | 68.30 |
| + Random | 60.00 | 58.85 | 60.16 | 70.00 | 65.73 | 64.69 | 71.33 | 69.98 | 70.05 |
| + GRESO | 56.67 | 58.33 | 57.24 | 69.17 | 71.77 | 69.79 | 69.89 | 70.96 | 70.19 |
| + PREPO (Ours) | 66.67 | 62.19 | 63.39 | 80.00 | 74.28 | 77.45 | **75.99** | **73.44** | **74.53** |

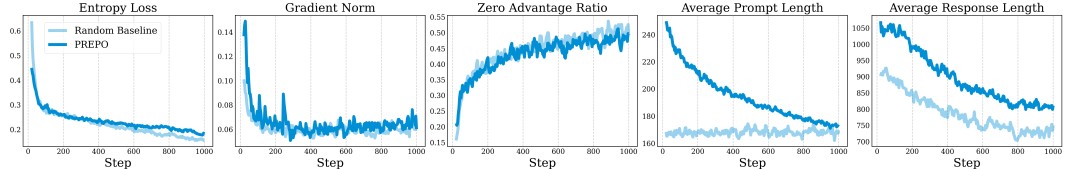

Figure 14: Full Comparison between PREPO and Random Selection on Qwen2.5-Math-1.5B.

Table 8: Performance Comparison (%) of PREPO with Different Selction Ratio ($K/\mathcal{B}$, $\mathcal{B}$ fixed) on Qwen2.5-Math-1.5B. *Best results are highlighted in bold or underlined.*

| Selection Ratio | AIME25 | AIME24 | MATH | Olympiad | Avg ↑ | # Rollouts ↓ |
|---|---|---|---|---|---|---|
| 30% | 20.00 | 20.83 | 75.85 | 25.00 | 35.42 | 1.4M |
| 25% | 13.33 | 20.00 | 73.55 | 24.00 | 32.72 | 1.5M |
| 20% | 20.00 | 16.67 | 76.25 | 32.00 | **36.23** | **1.1M** |
| 15% | 13.33 | 20.00 | 75.75 | 22.17 | 32.81 | 1.4M |
| 10% | 15.83 | 19.17 | 70.25 | 21.33 | 31.65 | 1.7M |
| 5% | 19.17 | 16.67 | 69.85 | 20.17 | 31.46 | 1.8M |

# H DISCUSSION

## H.1 RESULTS ON OTHER TRAINING DATASET

As DAPO-Math-17K contains both English and Chinese questions, prompt perplexity could correlate with language. To exclude the cofounder of language effects, we trained Qwen2.5-Math-7B on a purely English dataset selected from OpenR1-Math-220k[2], and compared PREPO with several baselines under the same training budget. The results are shown in Table A below. PREPO again achieves lower rollout usage while achieve a higher average performance. This indicates that the gains of PREPO are not driven by language imbalance.

Table 9: Comparison of PREPO and baselines trained on all-English data (Model: Qwen2.5-Math-7B)

| Method | AIME'25 | AIME'24 | MATH500 | Olympiad | Avg ↑ | #Rollout ↓ |
|--------|---------|---------|---------|----------|-------|------------|
| DS | 11.98 | 30.00 | 72.80 | 42.02 | 39.20 | 3256K |
| Random | 10.79 | 28.25 | 80.21 | 36.72 | 38.99 | 2457K |
| GRESO | 6.67 | 30.00 | 79.20 | 45.63 | 40.38 | 1331K |
| PREPO | 12.92 | 33.33 | 78.05 | 44.10 | **42.10** | **860K** |

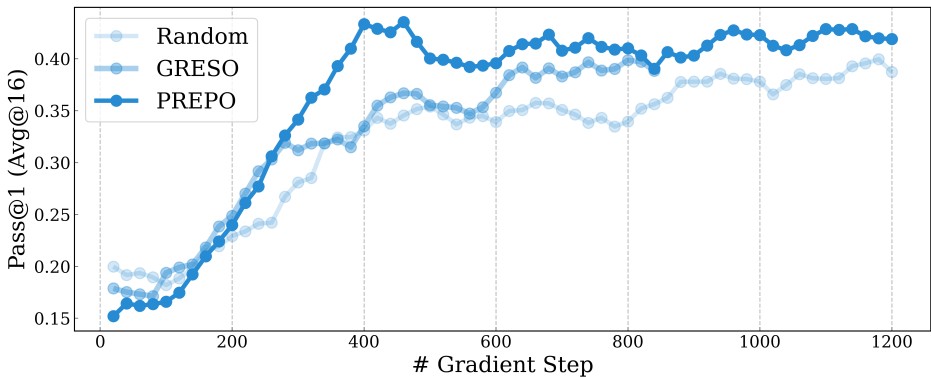

Figure 15: Comparison of performances across PREPO, GRESO, and Random (x-axis: training step, y-axis: validation score, model: Qwen2.5-Math-7B, training data: OpenR1-Math)

## H.2 WHAT DOES THE PPL-SCHEDULE CONTRIBUTE TO TRAINING?

We compared three configurations on Qwen2.5-Math-7B training exclusively with HIGH-PPL prompts, exclusively with LOW-PPL prompts, and the PPL-schedule, which gradually transitions from Low- to HIGH-PPL prompts. The training dynamics are shown in Figure 16.

In terms of entropy loss, the PPL-schedule achieves a balance between the two extremes: entropy decreases steadily but not excessively, thereby mitigating the risk of collapse. With respect to the zero-advantage ratio, the PPL-schedule consistently maintains a lower value, ensuring that a greater proportion of rollouts remain informative throughout training.

## H.3 WHAT DOES RELATIVE ENTROPY BRING TO TRAINING?

As shown in Figure 17, we found that PREPO with relative entropy could further reduce the zero-advantage ratio and thus improve sample efficiency.

**Case Analysis.** In Figure 18, we display examples of responses with their token-level entropy from the same mini-batch, where darker colors mean higher entropy, and each sequence is weighted by its relative entropy.

---

[2]https://huggingface.co/datasets/open-r1/OpenR1-Math-220k

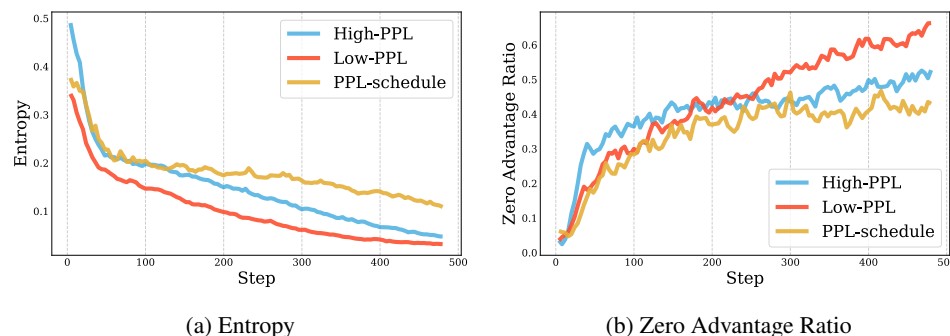

(a) Entropy                                    (b) Zero Advantage Ratio

Figure 16: Comparison of Training Dynamics between PPL-schedule and Static PPL Selection (Low- and HIGH-PPL groups)

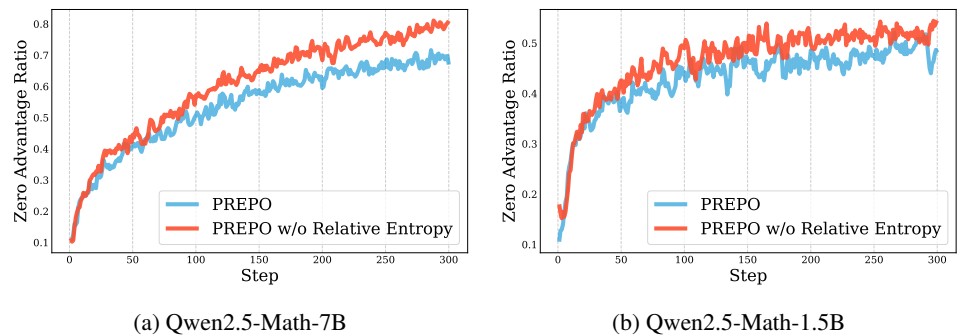

(a) Qwen2.5-Math-7B                            (b) Qwen2.5-Math-1.5B

Figure 17: Comparison of zero advantage ratio between PPL-schedule and PREPO.

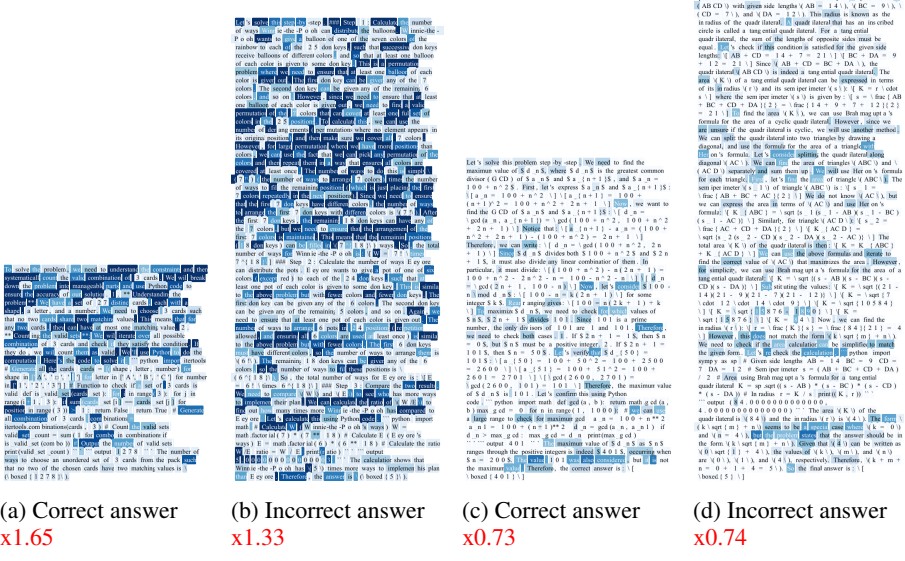

(a) Correct answer    (b) Incorrect answer    (c) Correct answer    (d) Incorrect answer
x1.65                 x1.33                    x0.73                 x0.74

Figure 18: Token-Level Entropy of Sequences within A Mini-batch (with Relative-Entropy in Red).

## H.4 DOES THE PROMPT PPL VARY DURING TRAINING?

As shown in Figure 19, we computed the range of PPL across at each training epoch and observed that it remained relatively stable throughout training. In addition, the PPL of the prompts used for training exhibited minimal variation.

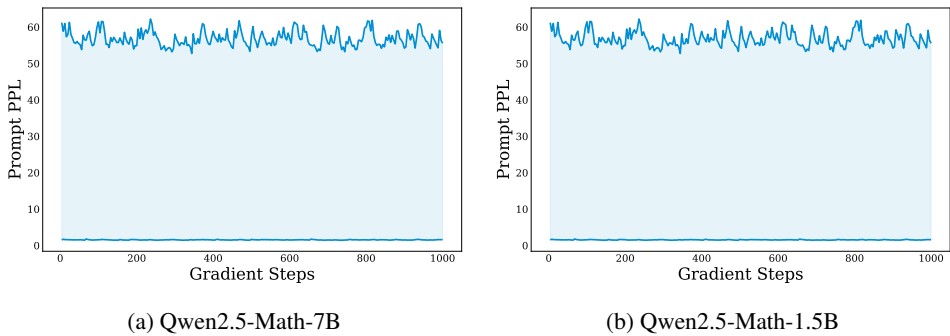

(a) Qwen2.5-Math-7B                (b) Qwen2.5-Math-1.5B

Figure 19: Range of prompt PPL during training.

### H.5 TIME CONSUMPTION OF PREPO

As shown in Figure 20, the time consumption of calculating prompt PPL is barely minimal compared to the rollout generation duration.

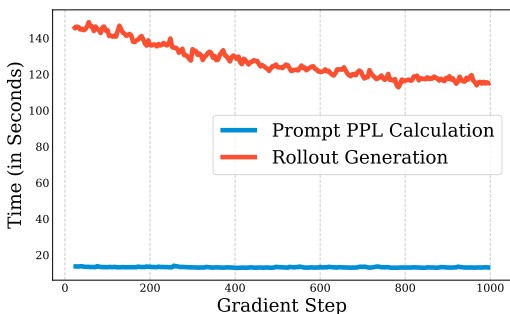

Figure 20: Comparison of calculating prompt PPL and rollout generation

### H.6 PROBLEM DIVERSITY IN PREPO

Analysis indicates that PREPO selects a more diverse set of problems compared to random sampling. Specifically, PREPO achieves broader coverage across Mathematics Subject Classification[3] (MSC) categories during training, as illustrated in Figure 21.

### H.7 DOES THE PREPO MODEL MEMORIZE THE TRAINING DATA?

Following Wu et al. (2025), we trimmed $40\%$ of each prompt to construct *partial problems*. We then evaluate models on these partial prompts and compute the average pass rate of 16 generations. As shown in Figure 22, the vast majority of partial problems have near-zero pass rate, with only a small fraction achieving non-trivial success. These distributions suggest that the model does not simply memorize training data, but instead requires the full problem context to solve tasks.

### H.8 SENSITIVITY TO EXTREME ENTROPIES

As shown in Figure 23, the distribution of relative entropy weights remains largely stable across training steps 100, 200, 300, and 400.

---

[3]MSC : https://zbmath.org/classification/

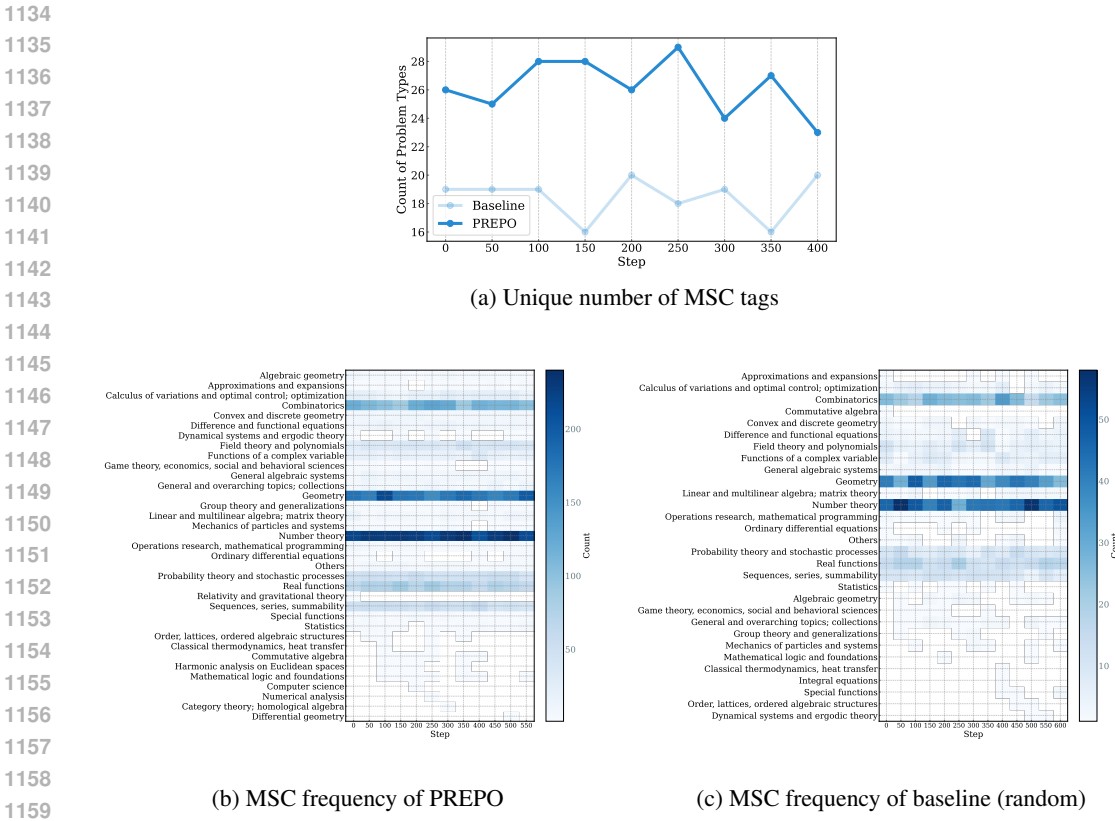

(a) Unique number of MSC tags

(b) MSC frequency of PREPO

(c) MSC frequency of baseline (random)

Figure 21: Comparison of problem diversity between PREPO and baseline on Qwen2.5-Math-7B.

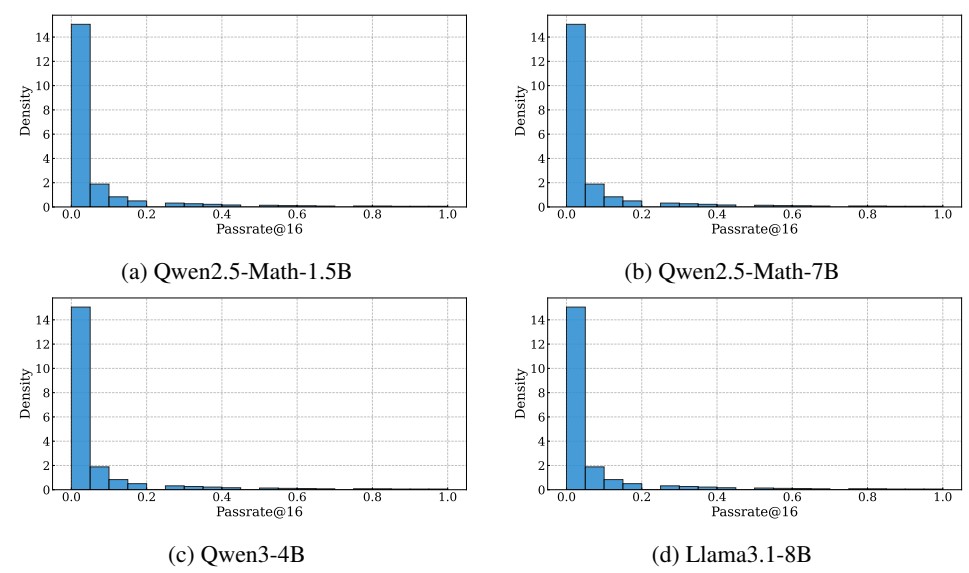

(a) Qwen2.5-Math-1.5B

(b) Qwen2.5-Math-7B

(c) Qwen3-4B

(d) Llama3.1-8B

Figure 22: Distribution of passrate@16 over all partial prompts

## H.9 COMPARISON WITH NO-FILTERING

For Qwen2.5-Math-7B, we observe that PREPO attains performance comparable to training on the full dataset without any filtering, i.e., using 5 times rollouts per step, as shown in Figure 24. The

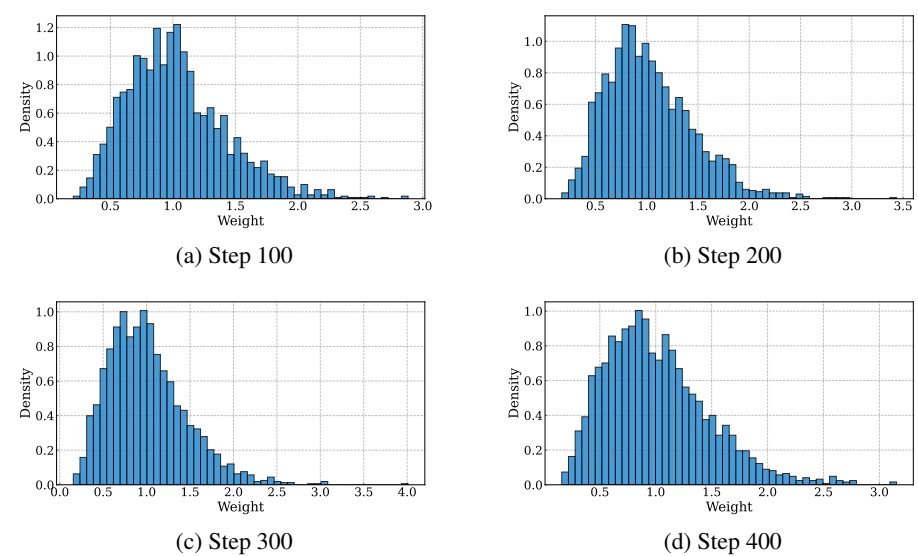

(a) Step 100

(b) Step 200

(c) Step 300

(d) Step 400

Figure 23: Weight Distribution During Training (Qwen2.5-Math-1.5B).

"Less is More" pattern implies that efficient selection can reduce the amount of data needed for RLVR training.

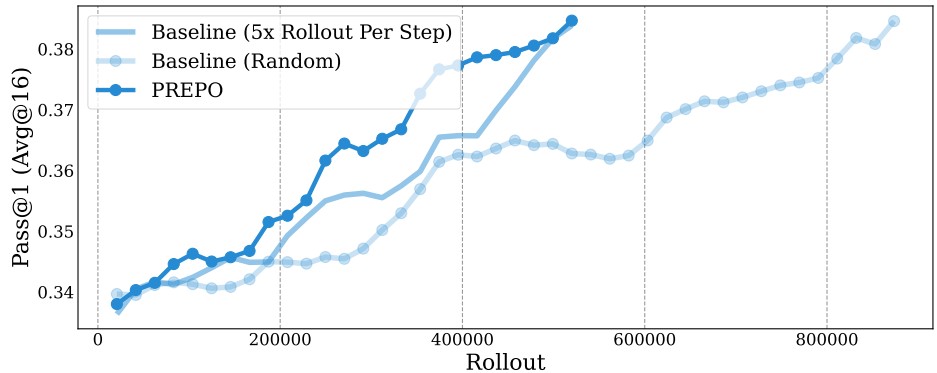

Figure 24: Comparison of PREPO and baselines (a) training w/o filtering (b) random selection.

