# OpenReview forum: "Interpretable Intrinsic Cues for Efficient Reinforcement Learning with Large Language Models"
_ICLR.cc/2026/Conference — ICLR 2026 Conference Withdrawn Submission_

### Official Review · Reviewer_1nMi · 2025-10-30

**Soundness:** 2
**Presentation:** 4
**Contribution:** 3
**Rating:** 4
**Confidence:** 3

**Summary:**

This paper introduces PREPO, a method for improving training efficiency in Reinforcement Learning with Verifiable Rewards (RLVR) through active sampling. The approach uses prompt perplexity as a measure of prompt difficulty, gradually shifting from easier to more challenging prompts during training.  In addition, it improves diversity in GRPO rollouts by prioritizing sequences with higher entropy, encouraging more exploratory and informative updates.

**Strengths:**

- The method is evaluated across multiple model families (Qwen and LLaMA), showing robustness and generality.
- The approach is clearly explained, with intuitive motivation and detailed algorithmic descriptions.
- PREPO achieves consistent rollout reduction across models while maintaining/improving accuracy.
- he method is simple yet effective, using interpretable intrinsic signals (perplexity and entropy) rather than external heuristics or external models.

**Weaknesses:**

- It appears that some relevant literature is missing: In particular [1] which it introduces CISPO an updated policy-gradient method with with truncated importance sampling. A discussion of how PREPO's entropy based weighting compares to CISPO would strengthen the paper.
- The reported baseline performance for some models (e.g., Qwen2.5-Math-7B) appears a lot lower than in prior work. Other studies, like [2](Table 3) have reported much higher base scores. This discrepancy raises concerns about the robustness of the evaluation setup. Some additional details on the evaluation procedure would help build confidence.
- For small benchmarks (like AIME'24) usually avg@32 or avg@64 is reported (see [2] or [3]). This is relevant because in Table 2 the performance is average across benchmarks and not number of samples. AIME'24 can have a variance of ~2% with avg@16. (This does not apply for MATH or Olympiad).

1: MiniMax-M1: Scaling Test-Time Compute Efficiently with Lightning Attention
2: A Sober Look at Progress in Language Model Reasoning: Pitfalls and Paths to Reproducibility
3: DeepSeek-R1: Incentivizing Reasoning Capability in LLMs via Reinforcement Learning

**Questions:**

- **Q1:** How stable is PREPO during extended training: Other methods like DAPO show training instabilities over longer training horizons. Does PREPO exhibit similar behavior?
- **Q2:** Evaluation
   - (a) Was a standardized evaluation framework used, or were the experiments based on a custom implementation?
   - (b) How was answer matching performed? E.g. by math-verify or an equivalent symbolic equivalence checker?

---

> ### Author Response · Authors · 2025-11-24
> **Response to Reviewer 1nMi (Part 1)**
>
> **W1** It appears that some relevant literature is missing: In particular [1] which it introduces CISPO an updated policy-gradient method with with truncated importance sampling. A discussion of how PREPO's entropy based weighting compares to CISPO would strengthen the paper.
>
> **A1** Thank you for pointing this out. We will add a discussion of CISPO to the related work section. CISPO addresses variance reduction in policy-gradient methods through truncated importance sampling, whereas PREPO focuses on online data selection and weighting based on model-driven perplexity and relative entropy. These two approaches target different aspects of RLVR training, and thus can be viewed as complementary rather than overlapping. We appreciate the suggestion and will clarify this connection in the revised manuscript.
>
> ---
>
> **W2** The reported baseline performance for some models (e.g., Qwen2.5-Math-7B) appears a lot lower than in prior work [before training]. Other studies, like [2](Table 3) have reported much higher base scores. This discrepancy raises concerns about the robustness of the evaluation setup. Some additional details on the evaluation procedure would help build confidence.
>
> **A2** Thank you for raising this point. We identified an inconsistency in the evaluation procedure for the base models and have corrected it to ensure alignment with the evaluation practices for the checkpoints during training. For all experiments, all model scores are computed using the same validation routine as the other checkpoints. Specifically, we now all use the `_validate()` function from the official VeRL implementation (link provided below), which standardizes decoding and scoring across runs:
>
> https://github.com/volcengine/verl/blob/fb532783ad3176b4f2a1acbe4f75a5d695b4e0b4/verl/trainer/ppo/ray_trainer.py\#L569

---

> ### Author Response · Authors · 2025-11-24
> **Response to Reviewer 1nMi (Part 2)**
>
> **W3** For small benchmarks (like AIME'24) usually avg@32 or avg@64 is reported (see [2] or [3]). This is relevant because in Table 2 the performance is average across benchmarks and not number of samples. AIME'24 can have a variance of ~2% with avg@16. (This does not apply for MATH or Olympiad).
>
> **A3** Thank you for the suggestion. We agree that AIMEs could have higher variance when evaluated with a small number of samples, and that reporting avg@32 or avg@64 provides a more stable estimate. Following your recommendation, we re-evaluated the AIME scores for the reported checkpoints using avg@16, avg@32, and avg@64. Table D shows the results for Qwen2.5-7B and Qwen2.5-Math-7B. Across all evaluation settings, PREPO continues to achieve the highest average performance.
>
> **Table D: Details of performance comparison (%) on Qwen models.** *Avg (avg@16/32/64) means the average of AIME25 (avg@16/32/64), AIME24 (avg@16/32/64), MATH, and Olympiad. Best results are highlighted in bold or underlined.*
>
> | Method | AIME25 (avg@16) | AIME25 (avg@32) | AIME25 (avg@64) | AIME24 (avg@16) | AIME24 (avg@32) | AIME24 (avg@64) | Avg (avg@16) | Avg (avg@32) | Avg (avg@64) |
> |--------|------------------|------------------|------------------|------------------|------------------|------------------|----------------|----------------|----------------|
> | **Qwen2.5-7B** | 1.25 | 2.60 | 1.17 | 4.17 | 4.90 | 4.95 | 27.69 | 28.21 | 27.87 |
> | + DS | 7.92 | 7.08 | 7.19 | 17.55 | 15.00 | 14.90 | 34.97 | 34.76 | 34.13 |
> | + Random | 6.98 | 6.35 | 6.51 | 16.41 | 15.63 | 15.89 | 34.39 | 34.04 | 34.14 |
> | + GRESO | 9.22 | 7.29 | 7.50 | 10.83 | 13.65 | 13.39 | 34.39 | 34.92 | 34.90 |
> | + PREPO (Ours) | **10.21** | **10.62** | **10.00** | **16.09** | **15.52** | **16.35** | **35.61** | **35.72** | **35.63** |
> | **Qwen2.5-Math-1.5B** | 3.54 | 3.75 | 4.48 | 10.21 | 8.33 | 8.96 | 24.23 | 23.81 | 24.15 |
> | + DS | 10.83 | 13.76 | 14.54 | 25.83 | 21.72 | **23.04** | 34.10 | 33.80 | 34.33 |
> | + Random | 20.00 | 17.45 | 18.81 | 16.67 | 15.42 | 16.50 | 35.86 | 34.91 | 35.52 |
> | + GRESO | 15.38 | 11.40 | 15.39 | 20.00 | 19.20 | 17.97 | 34.16 | 32.86 | 33.56 |
> | + PREPO (Ours) | **20.00** | **19.69** | **19.38** | **16.67** | **16.98** | 17.03 | **36.23** | **36.23** | **36.17** |
> | **Qwen2.5-Math-7B** | 8.95 | 9.17 | 8.80 | 17.50 | 20.80 | 19.22 | 35.45 | 35.45 | 34.96 |
> | + DS | 13.33 | 14.27 | 14.84 | 33.33 | 35.63 | 34.69 | 34.10 | 40.36 | 40.26 |
> | + Random | 10.00 | 12.50 | 12.50 | 26.67 | 29.34 | 29.87 | 35.86 | 40.73 | 40.86 |
> | + GRESO | 18.33 | 18.58 | 18.58 | 25.83 | 24.38 | 25.00 | 37.46 | 36.90 | 37.05 |
> | + PREPO (Ours) | **12.81** | **14.41** | **15.93** | **26.15** | **29.17** | **29.25** | **39.59** | **40.75** | **41.15** |
> | **Qwen3-4B** | 30.00 | 47.30 | 44.89 | 53.33 | 61.60 | 28.70 | 57.53 | 63.92 | 55.09 |
> | + DS | 63.33 | 53.90 | 54.80 | 66.67 | 63.50 | 65.30 | 70.78 | 67.63 | 68.10 |
> | + Random | 60.00 | 58.85 | 60.16 | 70.00 | 65.73 | 64.69 | 71.33 | 69.98 | 70.05 |
> | + GRESO | 56.67 | 58.33 | 57.24 | 69.17 | 71.77 | 69.79 | 69.89 | 70.96 | 70.19 |
> | + PREPO (Ours) | **66.67** | **62.19** | **63.39** | **80.00** | **74.28** | **77.45** | **75.99** | **73.44** | **74.53** |
>
> We have added the full set of comparisons for all models to the revised manuscript.

---

> > ### Author Response · Authors · 2025-11-24
> > **Response to Reviewer 1nMi (Part 3)**
> >
> > **Q1** How stable is PREPO during extended training: Other methods like DAPO show training instabilities over longer training horizons. Does PREPO exhibit similar behavior?
> >
> > **A1** We did not observe training instabilities with PREPO in our experiments, as shown in the performance curves in Figure 1 and Figure 15. One possible explanation is that the PPL-based selection changes smoothly over training, which reduces abrupt shifts in the sampled data.
> >
> > ---
> >
> > **Q2** (a) Was a standardized evaluation framework used, or were the experiments based on a custom implementation? (b) How was answer matching performed? E.g. by math-verify or an equivalent symbolic equivalence checker?
> >
> > **A2** (a) We use the `_validate()` function in the link below to evaluate the model.https://github.com/volcengine/verl/blob/fb532783ad3176b4f2a1acbe4f75a5d695b4e0b4/verl/trainer/ppo/ray_trainer.py#L569
> > (b) The basic verifier is `math-verify` and we add latex normalization package (`latex2sympy2_extended`) for AIME, and Olympiad bench.

---

> ### Comment · Reviewer_1nMi · 2025-11-26
> **Response to Authors' Rebuttal**
>
> I thank the authors for addressing my questions and for conducting additional experiments. Fixing the evaluation procedure is essential and adhering to established standards is important, so I am pleased that the authors now align with this. I will increase my score.

---

> > ### Author Response · Authors · 2025-11-26
> > **Response to Reviewer’s Increased Rating**
> >
> > We thank the reviewer for the positive feedback and for acknowledging the corrected evaluation procedure. We are glad the revisions addressed the concerns and appreciate the increased score.

---

### Official Review · Reviewer_oUFb · 2025-10-31

**Soundness:** 2
**Presentation:** 3
**Contribution:** 3
**Rating:** 4
**Confidence:** 4

**Summary:**

This paper proposes PREPO which consists of 2 contributions while doing RLVR

1. Using prompt perplexity as an implicit curriculum since they observe that prompt perplexity is correlated with hardness (measured using pass rate).
2. Amplifying diversity in rollouts using the relative entropy between rollouts by multiplying the policy gradient per rollout with a corresponding weight.

Using these, they show improvements on a bunch of base models such as Llama3.1, Qwen2.5 and Qwen3.

**Strengths:**

The idea of using prompt perplexity is novel. It is very useful since you don't need to do rollouts to estimate the value of each prompt and this significantly reduces the computational burden of inference. The paper is also written well and does a good number of experiments.

**Weaknesses:**

There are a couple of concerns with the experiments:

1. The authors should put the result of PREPO w/o relative entropy in Table 2 since it's unclear what contributes to the improvements in performance. Is performance improved because of the entropy weighting or is it because you select more informative prompts?

2. The authors should also do an experiment without the prompt filtering, just using the entropy weighting to ablate this component of their contribution.

3. The numbers in Table 2 are slightly unsettling. For instance, most of the gains for Qwen2.5-Math are coming from the Olympiad test set, gains for Qwen3 are coming from AIME and gains for Qwen2.5-7B are coming from AIME24. Does something explain these mixed gains?

4. Also, could the authors put validation performance on y-axis and time/rollouts on x-axis and show the curve for all methods to make the training dynamics clearer?

**Questions:**

Please look at the weaknesses.

---

> ### Author Response · Authors · 2025-11-26
>
> ## Reviewer oUFb
>
>
> **W1** The authors should put the result of PREPO w/o relative entropy in Table 2 since it's unclear what contributes to the improvements in performance. Is performance improved because of the entropy weighting, or is it because you select more informative prompts? The authors should also do an experiment without the prompt filtering, just using the entropy weighting to ablate this component of their contribution.
>
> **A1** Thanks for your comments. We have updated the ablation studies, including `w/o PPL-schedule` that reduces the method to random sampling plus relative entropy weighting.
>
> From the full ablations in Tables A and B below, we find that both components contribute to the overall effectiveness of PREPO. PREPO performs better than random sampling (w/o PPL-schedule) because it ranks prompts using the model’s own perplexity signal, producing a selection pattern that evolves smoothly as training progresses rather than sampling uniformly at all times.
>
>
> **Table A: Ablation study on Qwen.**
> | Model               | Method                    | AIME25 | AIME24 | MATH  | Olympiad Bench | Avg ↑ |
> |---------------------|----------------------------|--------|--------|-------|----------------|-------|
> | Qwen2.5-Math-7B     | PREPO                     | 12.81  | 26.15  | 77.80 | 41.58          | **39.59** |
> |                     | w/o relative entropy       | 10.00  | 23.33  | 74.60 | 39.21          | 36.79 |
> |                     | w/o PPL-schedule           | 11.87  | 25.73  | 76.48 | 34.43          | 37.13 |
> | Qwen2.5-7B          | PREPO                     | 10.20  | 16.09  | 76.30 | 39.85          | **35.61** |
> |                     | w/o relative entropy       | 6.98   | 16.41  | 75.70 | 38.47          | 34.39 |
> |                     | w/o PPL-schedule           | 8.89   | 16.04  | 79.41 | 22.54          | 31.72 |
> | Qwen2.5-Math-1.5B   | PREPO                     | 20.00  | 16.67  | 76.25 | 32.00          | **36.23** |
> |                     | w/o relative entropy       | 10.21  | 15.68  | 72.10 | 30.50          | 32.12 |
> |                     | w/o PPL-schedule           | 11.81  | 13.12  | 70.21 | 30.02          | 31.29 |
> | Qwen3-4B            | PREPO                     | 66.67  | 80.00  | 96.60 | 60.67          | **75.99** |
> |                     | w/o relative entropy       | 64.77  | 72.70  | 90.54 | 59.06          | 71.77 |
> |                     | w/o PPL-schedule           | 64.99  | 77.73  | 95.03 | 57.86 |  73.90
>
>
> **Table B: Ablation study on Llama.**
> | Model          | Method                  | GSM8K | MATH  | Avg ↑ |
> |----------------|--------------------------|-------|-------|-------|
> | Llama3.1-8B    | PREPO                   | 51.10 | 21.81 | **36.55** |
> |                | w/o relative entropy     | 46.85 | 18.25 | 32.55 |
> |                | w/o PPL-schedule         | 34.55 | 16.29 | 25.42 |
>
> ---
>
> **W2** The numbers in Table 2 are slightly unsettling. For instance, most of the gains for Qwen2.5-Math are coming from the Olympiad test set, gains for Qwen3 are coming from AIME and gains for Qwen2.5-7B are coming from AIME24. Does something explain these mixed gains?
>
> **A2** Thanks for making this interesting remark! The mixed gains arise probably because all models are trained on the same data (DAPO-Math-17K and MATH500), but their perplexity distributions and training dynamics differ. PREPO’s schedule depends on model-induced PPL, so each backbone moves through the DAPO-Math prompts at a different pace and emphasizes different parts of the training distribution. As a result, the improvements manifest on different evaluation subsets for different models.
>
> Importantly, the gains are not tied to a single benchmark. For example, for Qwen2.5-Math-1.5B, the improvement on AIME25 is large (3.54 → 20.00), not only on the Olympiad bench. Similar cross-benchmark improvements appear in the other models. This indicates that the effect is model-specific rather than dataset-specific, even though all models are trained on the same math data.
>
> ---
>
> **W3** Also, could the authors put validation performance on y-axis and time/rollouts on x-axis and show the curve for all methods to make the training dynamics clearer?
>
> **A3** Thanks for your suggestions! We have updated Figure 15 in the Appendix with a comparison of validation performance on the y-axis and training progress on the x-axis.

---

### Official Review · Reviewer_gvHC · 2025-11-07

**Soundness:** 3
**Presentation:** 4
**Contribution:** 3
**Rating:** 4
**Confidence:** 3

**Summary:**

The paper proposes PREPO, a lightweight reinforcement learning (RLVR) framework that improves data efficiency by leveraging intrinsic signals (perplexity and entropy) instead of heuristics or auxiliary models. The method is simple to implement and requires a negligible amount of additional computations while reducing the rollouts by 2-3 times compared to the Dynamic Sampling.

**Strengths:**

1. PREPO is extremely simple to implement. It only requires computing prompt perplexity and token-level entropy. The integration into standard PPO/RLVR loops is almost trivial.
2. Its computational overhead is negligible since all intrinsic cues (perplexity and entropy) are derived from existing training signals.
3. The paper is clearly written, empirically rich, and includes diagnostic analyses that help readers understand why and how the method improves efficiency.

**Weaknesses:**

1. **Oversimplification of the PPL-Schedule**: The paper's central curriculum strategy relies on a fixed linear schedule that shifts from low-PPL to high-PPL prompts. This feels arbitrary and rigid. I also wonder how authors chose hyperparameters (e.g., T, N, K). The value of T seems to determine both the total training steps and the transition rate. It might affect the number of rollouts and final performance critically.

2. **The PPL-Schedule's Conflicting Goals**: The "easy-to-hard" schedule seems to prioritize initial training speed. It might risk sacrificing peak performance by potentially under-sampling the more valuable (HIGH-PPL) data, especially in the critical later stages of training. The paper needs to justify why this efficiency-focused curriculum is superior to a performance-focused one that might, for example, heavily prioritize HIGH-PPL prompts.

3. **Conflation of "Difficulty" with "Domain"**: The paper equates low PPL with "easy" and high PPL with "hard". However, the analysis in Appendix E suggests an alternative explanation: HIGH-PPL prompts contain significantly more non-English characters. This implies PPL might be acting as a proxy for domain mismatch rather than pure task difficulty. The PPL-schedule could therefore be interpreted as a curriculum that trains first on in-domain data (English math) and then gradually introduces out-of-domain data (non-English math). This is a valid curriculum, but it's a different claim than "easy-to-hard" and has different implications for generalization.

4. **Missing Key Ablation Studies**: The paper's only ablation (Section 5.4) compares the full PREPO against just the PPL-schedule component. This merely shows that entropy weighting adds some benefit. *Missing Study 1*: One missing comparison is [Random Selection] + [Entropy Weighting]. Without this baseline, it is impossible to know how much of the final performance gain comes from the sophisticated PPL-schedule versus the (perhaps simpler) entropy weighting. If this baseline performed comparably to PREPO, it would suggest the entire PPL-schedule is unnecessary. *Missing Study 2*: The paper also needs to elaborate on the design of the PPL-schedule itself. The authors mention (Section 4.2) that other schedules are possible but provide no empirical justification for their choice of a linear schedule over other options or static baselines.

5. **Average Response Length (ARL) Increase and Inference Inefficiency**: According to the Fig. 5 and 14, PREPO increases ARL compared to the baseline. An increase in ARL can translate to an increase in total token generation cost (GPU time, energy consumption) per rollout. If the ARL is significantly longer for PREPO, the claimed reduction in rollout count may be offset when measuring the true cost by total tokens generated ($\text{Rollout Count} \times \text{Average Length}$). Furthermore, it can also lead to an inference inefficiency.

**Questions:**

1. Compared to the GRESO[1] paper's results, the total rollouts and the performance are quite different (e.g., Qwen2.5-Math-7B; DS: 13.1M vs 1.6M and GRESO: 6.3M vs 0.65M). Is this because the training budget is different? If so, what would happen if we increase the total training budget as much as GRESO did?

2. In Fig. 22, the baseline with 5x rollout per step shows comparable performance with PREPO. In this case, what is the advantage of PREPO compared to the baseline with 5x rollout per step?

3. In Tab. 6, why low selection ratio show more rollouts? At each training step, don't we use a smaller number of samples to generate the rollouts?

4. In Fig. 5 and 14, why does the Average Prompt Length decrease?

5. In Sec 5.5, it is mentioned that further discussion about the sensitivity of large entropy can be found in Appendix H, but I cannot find one (I might have missed something). Can you elaborate more on how the distribution of weights changes during the overall training?


[1] Haizhong Zheng, Yang Zhou, Brian R Bartoldson, Bhavya Kailkhura, Fan Lai, Jiawei Zhao, and Beidi Chen. Act only when it pays: Efficient reinforcement learning for llm reasoning via selective rollouts. arXiv preprint arXiv:2506.02177, 2025.

---

> ### Author Response · Authors · 2025-11-24
> **Response to Reviewer gvHC (Part 1)**
>
> **W1** The paper's central curriculum strategy relies on a fixed linear schedule that shifts from low-PPL to high-PPL prompts. This feels arbitrary and rigid. I also wonder how authors chose hyperparameters (e.g., T, N, K). The value of T seems to determine both the total training steps and the transition rate. It might affect the number of rollouts and final performance critically.
>
> **A1** Thank you for raising this point. Although the schedule is expressed in a linear form, it is not rigid. The actual ordering is determined entirely by the model’s own perplexity estimates at each step. This makes the curriculum adaptive to both the model and the dataset, rather than a fixed or hand-crafted notion of difficulty.
>
> We set $T = 1000$ in all experiments. This is the maximum number of training steps allowed. In our evaluation, we do not use all $T$ steps when reporting results. Instead, following prior work, we evaluate every 50 steps and report the performance at the step where the model achieves its best score. This often occurs well before step $T$.
>
> Because the schedule is normalized by training progress and relies only on model-induced perplexity, it remains robust when scaling $T$, $N$, $K$. If compute resources increase, these quantities can be scaled proportionally and the same ratio $N/K$.
>
> ---
>
> **W2** The "easy-to-hard" schedule seems to prioritize initial training speed. It might risk sacrificing peak performance by potentially under-sampling the more valuable (HIGH-PPL) data, especially in the critical later stages of training. The paper needs to justify why this efficiency-focused curriculum is superior to a performance-focused one that might, for example, heavily prioritize HIGH-PPL prompts.
>
> **A2** Thank you for the comment. The goal of the PPL-schedule is not only to speed up the early phase of training, but also to maintain strong performance in the later phase. Our preliminary analysis shows that low-PPL data tends to produce higher rewards and higher validation scores at the beginning, but it also increases the zero-advantage ratio if used exclusively. For this reason, the schedule gradually shifts toward higher-PPL prompts as training progresses.
>
> This design directly addresses your concern. The later stage of training samples a larger portion of the high-PPL data, which allows the model to benefit from the harder prompts without relying on them too heavily at the start, when they are less useful. In this way, the schedule supports both goals: it improves efficiency in the early stage and preserves strong final performance.
>
> We therefore do not view efficiency-focused and performance-focused training as competing objectives in this setting. The schedule is constructed to support both, and our experiments confirm that the model maintains its performance while gaining speed. We appreciate your attention to this point and hope our explanation can address your concern.
>
> ---
>
> **W3** The paper equates low PPL with "easy" and high PPL with "hard". However, the analysis in Appendix E suggests an alternative explanation: HIGH-PPL prompts contain significantly more non-English characters. This implies PPL might be acting as a proxy for domain mismatch rather than pure task difficulty. The PPL-schedule could therefore be interpreted as a curriculum that trains first on in-domain data (English math) and then gradually introduces out-of-domain data (non-English math). This is a valid curriculum, but it's a different claim than "easy-to-hard" and has different implications for generalization.
>
> **A3** Thank you for the comment. The preliminary results in Section 3.1 were intended to show a correlation between prompt perplexity and existing data selection metrics, not to define prompt perplexity as a direct measure of difficulty. Prompt perplexity captures how well the model predicts the input distribution, and in this sense, the PPL-schedule functions as an online selection method that moves from more in-domain prompts to less in-domain prompts. This matches your interpretation. We appreciate your comment and have revised Section 4.2 to make this distinction explicit and to avoid any confusion.

---

> > ### Author Response · Authors · 2025-11-24
> > **Response to Reviewer gvHC (Part 2)**
> >
> > **W4** The paper's only ablation (Section 5.4) compares the full PREPO against just the PPL-schedule component. This merely shows that entropy weighting adds some benefit. _Missing Study 1_: One missing comparison is [Random Selection] + [Entropy Weighting]. Without this baseline, it is impossible to know how much of the final performance gain comes from the sophisticated PPL-schedule versus the (perhaps simpler) entropy weighting. If this baseline performed comparably to PREPO, it would suggest the entire PPL-schedule is unnecessary. _Missing Study 2_: The paper also needs to elaborate on the design of the PPL-schedule itself. The authors mention (Section 4.2) that other schedules are possible but provide no empirical justification for their choice of a linear schedule over other options or static baselines.
> >
> > **A4** Thank you for these helpful suggestions. We have incorporated both requested ablations into the revised manuscript. (1) [Random Selection] + [Entropy Weighting], which isolates the contribution of the PPL-schedule, and
> > (2) non-linear variants of the PPL-schedule, which examine whether the linear form is essential.
> >
> >
> > For the non-linear schedule, we modify the linear progression by raising the normalized training progress $\rho$ to a power $\alpha$. Thus, the general schedule form is $l(\rho) = \big\lfloor \rho^{\alpha} \cdot (N - K) \big\rfloor$. We set $\alpha = 0.5$ in the ablation experiments, which causes the shift toward higher-PPL prompts to occur earlier in training.
> >
> > The results are summarized in Table A and Table B below. The [Random Selection] + [Entropy Weighting] baseline performs consistently below PREPO across all model sizes. This indicates that entropy weighting alone does not account for the gains observed with PREPO. The non-linear schedule performs above the static baselines but remains slightly below the linear schedule. This suggests that the linear schedule is already an effective choice and does not require a more complex design to achieve strong performance.
> >
> >
> > **Table A: Ablation study on Qwen. Best results are in bold. (w/o PPL-schedule = [Random Selection] + [Entropy Weighting])**
> > | Model               | Method                    | AIME25 | AIME24 | MATH  | Olympiad Bench | Avg ↑ |
> > |---------------------|----------------------------|--------|--------|-------|----------------|-------|
> > | Qwen2.5-Math-7B     | PREPO                     | 12.81  | 26.15  | 77.80 | 41.58          | **39.59** |
> > |                     | w/o PPL-schedule           | 11.87  | 25.73  | 76.48 | 34.43          | 37.13 |
> > |                     | w/ non-linear schedule     | 12.71  | 26.15  | 76.40 | 40.12          | 38.85 |
> > | Qwen2.5-7B          | PREPO                     | 10.20  | 16.09  | 76.30 | 39.85          | **35.61** |
> > |                     | w/o PPL-schedule           | 8.89   | 16.04  | 79.41 | 22.54          | 31.72 |
> > |                     | w/ non-linear schedule     | 9.58   | 16.25  | 76.31 | 39.60          | 35.44 |
> > | Qwen2.5-Math-1.5B   | PREPO                     | 20.00  | 16.67  | 76.25 | 32.00          | **36.23** |
> > |                     | w/o PPL-schedule           | 11.81  | 13.12  | 70.21 | 30.02          | 31.29 |
> > |                     | w/ non-linear schedule     | 9.15   | 15.83  | 75.85 | 30.26          | 32.77 |
> > | Qwen3-4B            | PREPO                     | 66.67  | 80.00  | 96.60 | 60.67          | **75.99** |
> > |                     | w/o PPL-schedule         | 64.99  | 77.73  | 95.03 | 57.86 |  73.90 |
> > |                     | w/ non-linear schedule           | 61.45  | 74.47  | 92.37 | 60.67          | 72.24 |
> >
> >
> > **Table B: Ablation study on Llama. Best results are in bold. (w/o PPL-schedule = [Random Selection] + [Entropy Weighting])**
> > | Model          | Method                  | GSM8K | MATH  | Avg ↑ |
> > |----------------|--------------------------|-------|-------|-------|
> > | Llama3.1-8B    | PREPO                   | 51.10 | 21.81 | **36.55** |
> > |                | w/o PPL-schedule         | 34.55 | 16.29 | 25.42 |
> > |                | w/ non-linear schedule   | 46.85 | 20.56 | 33.71 |

---

> > > ### Author Response · Authors · 2025-11-24
> > > **Response to Reviewer gvHC (Part 3)**
> > >
> > > **W5** Average Response Length (ARL) Increase and Inference Inefficiency: According to the Fig. 5 and 14, PREPO increases ARL compared to the baseline. An increase in ARL can translate to an increase in total token generation cost (GPU time, energy consumption) per rollout. If the ARL is significantly longer for PREPO, the claimed reduction in rollout count may be offset when measuring the true cost by total tokens generated (Token count). Furthermore, it can also lead to an inference inefficiency.
> > >
> > > **A5** Thanks for bringing out this point. We noticed that there is indeed a small difference betweeen the average response length bewteen PRPO and random selection baseline. However, we found the total generated tokens are always at the same scale.
> > >
> > > For Fig. 5, we provide the comparison of token counts between PREPO and random selection baseline, as shown in the table below.
> > >
> > > | Step | Random Token Counts|PREPO Token Counts|
> > > |------|--------------------|------------------|
> > > | 100  | 6,901,760          | 7,188,480        |
> > > | 500  | 16,281,600         | 16,896,000       |
> > > | 1000 | 24,760,320         | 26,337,280       |
> > > | 1500 | 33,421,440         | 35,841,920       |
> > >
> > >
> > > ---
> > >
> > > **Q1** Compared to the GRESO[1] paper's results, the total rollouts and the performance are quite different (e.g., Qwen2.5-Math-7B; DS: 13.1M vs 1.6M and GRESO: 6.3M vs 0.65M). Is this because the training budget is different? If so, what would happen if we increase the total training budget as much as GRESO did?
> > >
> > > **A1** We set clip-higher ratio as 0.28 to encourage the model can update those low-probability tokens and facilitates the generation of more diverse samples. By learning from more diverse rollouts, the model can perform on the validation faster.
> > >
> > > ---
> > >
> > > **Q2** In Fig. 22, the baseline with 5x rollout per step shows comparable performance with PREPO. In this case, what is the advantage of PREPO compared to the baseline with 5x rollout per step?
> > >
> > > **A2** PREPO reaches comparable performance while using only one-fifth of the rollout budget. In other words, the baseline needs 5× more rollouts to match the accuracy that PREPO attains with much less data. This aligns with prior observations that efficient selection can reduce the amount of data needed for RLVR training [1].
> > >
> > > [1] Ye, Y., Huang, Z., Xiao, Y., Chern, E., Xia, S., & Liu, P. (2025). LIMO: Less is More for Reasoning. COLM 2025.
> > >
> > > ---
> > >
> > > **Q3** In Tab. 6, why low selection ratio show more rollouts? At each training step, don't we use a smaller number of samples to generate the rollouts?
> > >
> > > **A3** A low selection ratio indeed leads to less number of samples to generate the rollouts, but it would require more training steps to achieve its best performance.
> > >
> > > ---
> > >
> > > **Q4** In Fig. 5 and 14, why does the Average Prompt Length decrease?
> > >
> > > **A4** Because we shift towards training data with the higher-PPL prompts.
> > >
> > > ---
> > >
> > > **Q5** In Sec 5.5, it is mentioned that further discussion about the sensitivity of large entropy can be found in Appendix H, but I cannot find one (I might have missed something). Can you elaborate more on how the distribution of weights changes during the overall training?
> > >
> > > **A5** Thank your careful reading.  The distribution of weights does not change much during training. We have included the weight distribution at the steps 100, 200, 300, and 400 in Appendix H.

---

### Official Review · Reviewer_Lz9P · 2025-11-07

**Soundness:** 3
**Presentation:** 3
**Contribution:** 2
**Rating:** 4
**Confidence:** 3

**Summary:**

This paper introduces the PREPO algorithm for RLVR for LLMs. The first idea is to use prompt perplexity as a proxity for problem difficulty, adopting a schedule from easy to challenging prompts. The second idea is to amplify diversity among rollouts by prioritizing sequences with greater emtropy. The algorithm achieves 2-3 times rollout reduction comparing to baselines.

**Strengths:**

- The paper proposes to use perplexity as a metric for problem difficulty. Comparing to dynamic sampling and random selection, perplexity-based sample selection method does not waste rollouts.
- Entropy-weighting can prevent the model from entropy collapse.

**Weaknesses:**

- The PPL-schedule-filter requires a schedule that is consistent with the learning dynamics of the model. If changing a model or a task, the schedule need to be tuned. Such a tuning of schedule introduces additional computational overhead not accounted for in the reported comparisons.
- The paper does not compare entropy-weighting with entropy-regularization methods. While it argues that entropy-weighting avoids additional hyperparameters, it is unclear whether the “hyperparameter-free” property provides meaningful practical advantages.

**Questions:**

See weakness

---

> ### Author Response · Authors · 2025-11-24
> **Response to Reviewer Lz9P (Part 1)**
>
> **W1** The PPL-schedule-filter requires a schedule that is consistent with the learning dynamics of the model. If changing a model or a task, the schedule need to be tuned. Such a tuning of schedule introduces additional computational overhead not accounted for in the reported comparisons.
>
> **A1** Thank you for the comment. We agree that it would be undesirable if the schedule required task-specific or model-specific tuning. Fortunately, the PPL-schedule is designed so that it does not need any manual adjustment. The schedule is computed directly from the model’s own perplexity estimates during training. This allows the method to adapt automatically when changing either the model or the dataset.
>
> The key point is that the schedule follows the training progress rather than any fixed set of hyperparameters. At each global step, the algorithm computes perplexity on the current batch of prompts and selects the subset that matches the current progress ratio. This removes the need for tuning and avoids extra computational cost. The implementation is simple and aligns with the Equations (2) (3) and (4) in the manuscript.
>
> ```python
> # compute perplexity for each prompt
> batch_ppl = self.actor_rollout_wg.get_ppl_batch(batch)
>
> # compute training progress
> rho = t / T
>
> # determine the segment of prompts to select
> start_idx = math.floor((N - K) * rho)
> indices = torch.argsort(batch_ppl['ppl'])[start_idx : start_idx + K]
>
> # select prompts based on the indices
> batch = batch.select_via_index(indices)
> ```
>
> This procedure uses only a few lines of code, requires no manual tuning, and adds no overhead beyond the perplexity computation that is already part of the rollout pipeline. We hope this clarification addresses your concern.

---

> > ### Author Response · Authors · 2025-11-24
> > **Response to Reviewer Lz9P (Part 2)**
> >
> > **W2** The paper does not compare entropy-weighting with entropy-regularization methods. While it argues that entropy-weighting avoids additional hyperparameters, it is unclear whether the “hyperparameter-free” property provides meaningful practical advantages.
> >
> > **A2** Thank you for raising this point. To address it, we added ablation experiments across all model scales. In these runs, we removed the entropy-weighting component in PREPO and replaced it with an entropy-regularization loss using a coefficient of 1e-3. Across all models, entropy-regularization either reduced downstream accuracy or caused an instability in the entropy term itself. In contrast, PREPO remained stable and achieved stronger results under the same training setup. These findings show that avoiding a manually tuned entropy coefficient does bring a practical benefit in RLVR training.
> >
> >
> > **Table A: Ablation study on Qwen (w/entropy loss means entropy-regularization with coefficient of 1e-3). We bold the best scores.**
> > | Model               | Method                    | AIME25 | AIME24 | MATH  | Olympiad Bench | Avg ↑ |
> > |---------------------|----------------------------|--------|--------|-------|----------------|-------|
> > | Qwen2.5-Math-7B     | PREPO                     | 12.81  | 26.15  | 77.80 | 41.58          | **39.59** |
> > |                     | w/ entropy loss            | 10.73  | 23.54  | 75.25 | 38.81          | 37.08 |
> > | Qwen2.5-7B          | PREPO                     | 10.20  | 16.09  | 76.30 | 39.85          | **35.61** |
> > |                     | w/ entropy loss            | 6.25   | 16.35  | 77.36 | 21.04          | 30.25 |
> > | Qwen2.5-Math-1.5B   | PREPO                     | 20.00  | 16.67  | 76.25 | 32.00          | **36.23** |
> > |                     | w/ entropy loss            | 6.46   | 16.56  | 73.77 | 30.99          | 31.95 |
> > | Qwen3-4B            | PREPO                     | 66.67  | 80.00  | 96.60 | 60.67          | **75.99** |
> > |                     | w/ entropy loss            | 61.10  | 75.17  | 88.39 | 60.54          | 71.28 |
> >
> > **Table B: Ablation study on Llama (w/entropy loss means entropy-regularization with coefficient of 1e-3). We bold the best scores.**
> > | Model          | Method                  | GSM8K | MATH  | Avg ↑ |
> > |----------------|--------------------------|-------|-------|-------|
> > | Llama3.1-8B    | PREPO                   | 51.10 | 21.81 | **36.55** |
> > |                | w/ entropy loss          | 4.45  | 8.30  | 6.33 |
> >
> > For the Llama model, the entropy-regularization run showed clear instability, with the entropy term increasing uncontrollably. PREPO did not show this behavior and remained stable throughout training.

---

### Official Review · Reviewer_t6D8 · 2025-11-12

**Soundness:** 2
**Presentation:** 3
**Contribution:** 3
**Rating:** 4
**Confidence:** 3

**Summary:**

This paper introduces two techniques in Reinforcement Learning with Verifiable Reward (RLVR) and proposes PREPO to accelerate the RLVR training. The first technique is an online curriculum learning trick which uses the prompt ppl as a proxy filtering criterion to sort the prompts, aiming at training the model on easy-to-hard questions in order. However, they found that easy questions (low-ppl ones) have more consistent rollouts and lead to fast entropy collapse, so they introduced the second trick: weighting the advantage functions by the relative sequence level entropy.

They show in experiments that PREPO achieves, on average, higher scores on benchmarks like AIME, MATH, and Olympiad with fewer rollouts. They show that to achieve similar or better performance, PREPO needs 2 to 3 times fewer rollouts. They also did ablation study on the two tricks they proposed.

**Strengths:**

1. The presentation in this submission is good, and the motivation is clearly stated. They start with a correlation analysis between the prompt ppl and the pass rate, and a comparison between the training dynamics on the low-ppl and high-ppl prompts. This clearly verifies the reason why they chose prompt ppl as the proxy criterion for the prompt difficulty and use it as the filtering criterion.

2. The main experiments showed that the method they proposed can save many rollouts to achieve similar performance. They compare PREPO with random selection, DAPO, and GRESO, and show that their method outperforms others in four benchmarks. The rollouts that they can save are striking.

**Weaknesses:**

1. Possibilities of other proxies for curriculum learning: In their submission, there are at least two alternative 'intrinsic data properties' that one can use to do curriculum learning. A: In their appendix E, they show that the low-ppl prompts usually have higher rates of English words and high-ppl ones often feature by non-English questions. Since they use DAPO-17k as the training set and it includes many Chinese questions, the non-English prompts will be an issue. I think one way to exclude this confounder one can use a purely English dataset as the training set. B: In Figure 5, they show that the average prompt lengths decrease (almost monotonically) in PREPO. This shows that when we rank the prompts by prompt ppl, we are always prioritizing the long prompts over shorter prompts. Does this mean we can simply use the prompts' length as a filtering criterion for the curriculum learning?

2. Issues on the Main Experiments:

A: I think they should add the two ablation studies in their Table 2 (the table for their main experiments). They should at least try the one without entropy weighting, and the one with entropy weighting but without the online filtering based on prompts ppl.

B: How did the authors choose the stopping time (how many rollouts they use in the main experiments?) For example, on the first block of their table 2, they show that GRESO uses 680k rollouts and PRESO uses 304k, and I wonder how they chose these two numbers? What happens if we train both methods (and all other methods) to the points where we use 1040k rollouts? I think a better way for comparison is to compare the (best) performance at the same rollouts. A better visualization way is to plot the curve for the average score (otherwise, we cannot exclude the hypothesis that the PRESO can be worse in the later training phase, and they finally converge to the same performance as GRESO or other baselines).

**Questions:**

See above.

**Details Of Ethics Concerns:**

/

---

> ### Author Response · Authors · 2025-11-24
> **Response to Reviewer t6D8 (Part 1)**
>
> **W1** Possibilities of other proxies for curriculum learning: In their submission, there are at least two alternative 'intrinsic data properties' that one can use to do curriculum learning. A: In their appendix E, they show that the low-ppl prompts usually have higher rates of English words and high-ppl ones often feature by non-English questions. Since they use DAPO-17k as the training set and it includes many Chinese questions, the non-English prompts will be an issue. I think one way to exclude this confounder one can use a purely English dataset as the training set. B: In Figure 5, they show that the average prompt lengths decrease (almost monotonically) in PREPO. This shows that when we rank the prompts by prompt ppl, we are always prioritizing the long prompts over shorter prompts. Does this mean we can simply use the prompts' length as a filtering criterion for the curriculum learning?
>
> **A1** Thank you for raising the question about alternative intrinsic data signals. We address both parts of the comment below.
>
> Regarding language effects (Point A): DAPO-Math-17K contains both English and Chinese questions, so prompt perplexity could correlate with language. To examine this directly, we trained Qwen2.5-Math-7B on a purely English dataset from OpenR1-Math, and compared PREPO with several baselines under the same training budget. The results are shown in Table A below. PREPO again achieves lower rollout usage while matching or exceeding downstream performance. This indicates that the gains are not driven by language imbalance. We have added these results and the associated validation curves to the revised manuscript.
>
> **Table A: Comparison of PREPO and baselines trained on all english dataset OpenR1-Math (Model: Qwen2.5-Math-7B)**
>
> | Method | AIME'25 | AIME'24 | MATH500 | Olympiad | Avg   | #Rollout |
> | ------ | ------- | ------- | ------- | -------- | ----- | -------- |
> | DS     | 11.98   | 30.00   | 72.80   | 42.02    | 39.20 | 3256K    |
> | Random | 10.79   | 28.25   | **80.21**   | 36.72    | 38.99 | 2457K    |
> | GRESO  | 6.67    | 30.00   | 79.20   | **45.63**   | 40.38 | 1331K    |
> | PREPO  | **12.92**   | **33.33**  | 78.05   | 44.10    | **42.10** | **860K**     |
>
> Regarding prompt length (Point B): Figure 5 shows a trend in the average prompt length under the PPL-schedule, but this does not mean that perplexity is determined by length. Perplexity measures the model’s probability over the entire sequence, whereas length is only a structural property. The two can correlate because longer prompts may contain more familiar patterns, but conceptually, perplexity and length do capture different aspects of the data. We use perplexity because it reflects the model’s own view of the inputs during training, though length is also a reasonable intrinsic property and may be worth examining in future work.

---

> > ### Author Response · Authors · 2025-11-24
> > **Response to Reviewer t6D8 (Part 2)**
> >
> > **W2** Issues on the Main Experiments: A: I think they should add the two ablation studies in their Table 2 (the table for their main experiments). They should at least try the one without entropy weighting, and the one with entropy weighting but without the online filtering based on prompts ppl.
> >
> > **A2** Thank you for the valuable suggestion. We have added both settings: (1) PREPO without entropy weighting, and (2) PREPO with entropy weighting but without the PPL-based online filtering. These results are now included in the revised manuscript.
> >
> > For your convenience, we also include the updated tables below. Table B and Table C show a consistent pattern across all tested model sizes, removing either component lowers the best average performance under the same maximum training steps. This indicates that the two components contribute in different ways.
> >
> > **Table B: Ablation study on Qwen.**
> > | Model               | Method                    | AIME25 | AIME24 | MATH  | Olympiad Bench | Avg ↑ |
> > |---------------------|----------------------------|--------|--------|-------|----------------|-------|
> > | Qwen2.5-Math-7B     | PREPO                     | 12.81  | 26.15  | 77.80 | 41.58          | **39.59** |
> > |                     | w/o relative entropy       | 10.00  | 23.33  | 74.60 | 39.21          | 36.79 |
> > |                     | w/o PPL-schedule           | 11.87  | 25.73  | 76.48 | 34.43          | 37.13 |
> > | Qwen2.5-7B          | PREPO                     | 10.20  | 16.09  | 76.30 | 39.85          | **35.61** |
> > |                     | w/o relative entropy       | 6.98   | 16.41  | 75.70 | 38.47          | 34.39 |
> > |                     | w/o PPL-schedule           | 8.89   | 16.04  | 79.41 | 22.54          | 31.72 |
> > | Qwen2.5-Math-1.5B   | PREPO                     | 20.00  | 16.67  | 76.25 | 32.00          | **36.23** |
> > |                     | w/o relative entropy       | 10.21  | 15.68  | 72.10 | 30.50          | 32.12 |
> > |                     | w/o PPL-schedule           | 11.81  | 13.12  | 70.21 | 30.02          | 31.29 |
> > | Qwen3-4B            | PREPO                     | 66.67  | 80.00  | 96.60 | 60.67          | **75.99** |
> > |                     | w/o relative entropy       | 64.77  | 72.70  | 90.54 | 59.06          | 71.77 |
> > |                     | w/o PPL-schedule           | 64.99  | 77.73  | 95.03 | 57.86 |  73.90 |
> >
> >
> > **Table C: Ablation study on Llama.**
> > | Model          | Method                  | GSM8K | MATH  | Avg ↑ |
> > |----------------|--------------------------|-------|-------|-------|
> > | Llama3.1-8B    | PREPO                   | 51.10 | 21.81 | **36.55** |
> > |                | w/o relative entropy     | 46.85 | 18.25 | 32.55 |
> > |                | w/o PPL-schedule         | 34.55 | 16.29 | 25.42 |
> >
> > PREPO performs better than random sampling (w/o PPL-schedule) because it ranks prompts using the model’s own perplexity signal, producing a selection pattern that evolves smoothly as training progresses rather than sampling uniformly at all times.
> >
> > PREPO performs better than w/o relative entropy because relative-entropy weighting influences how rollouts contribute to the update. Without it, low-entropy rollouts receive relatively higher weight, which changes the training signal and corresponds to weaker performance in our ablations.

---

> > > ### Author Response · Authors · 2025-11-24
> > > **Response to Reviewer t6D8 (Part 3)**
> > >
> > > **W3** How did the authors choose the stopping time (how many rollouts they use in the main experiments?) For example, on the first block of their table 2, they show that GRESO uses 680k rollouts and PRESO uses 304k, and I wonder how they chose these two numbers? What happens if we train both methods (and all other methods) to the points where we use 1040k rollouts? I think a better way for comparison is to compare the (best) performance at the same rollouts. A better visualization way is to plot the curve for the average score (otherwise, we cannot exclude the hypothesis that the PRESO can be worse in the later training phase, and they finally converge to the same performance as GRESO or other baselines).
> > >
> > >
> > > **A3** Thank you for raising this question about stopping criteria and rollout budgets. In all main experiments, we followed the standard practice used in prior work [1]. We evaluate the model every 50 steps and report the best average score reached during training. We have now stated this explicitly in Section 5.1 of the revised manuscript.
> > >
> > > In your example about different rollout counts (such as 680k vs. 304k), these numbers differ because each method reaches its best score at a different point in training. Since our goal is to study training efficiency, we compare methods based on how quickly they reach their highest performance rather than forcing all methods to use an identical number of rollouts after they have already stabilized. This evaluation is consistent across all baselines.
> > >
> > > For your concerns for the performance in later runs, we found that PREPO maintains its performance level and does not show a notable degradation in the later steps. We have added the Figure 1 and Figure 15 in the revised manuscript.
> > >
> > > [1] Haizhong Zheng, Yang Zhou, Brian R Bartoldson, Bhavya Kailkhura, Fan Lai, Jiawei Zhao, and Beidi Chen. Act only when it pays: Efficient reinforcement learning for llm reasoning via selective rollouts. arXiv preprint arXiv:2506.02177, 2025.

---

### Author Response · Authors · 2025-11-26
**Global Response**

We thank the reviewers for their careful reading and constructive feedback. The revisions improve the clarity of the presentation, broaden the experimental analysis, and better articulate the central ideas of the method. In response to the reviews,

1. We added all requested ablations, including PREPO without entropy weighting, PREPO with entropy weighting but without the PPL-schedule, nonlinear PPL-schedules, and entropy-regularization baselines.

2. We added cross-dataset and cross-model validation, experiments on a purely English dataset, updated AIME avg@32 and avg@64 metrics, and extended training-curve comparisons to address stability concerns.

3. We corrected an inconsistency affecting a subset of base model scores. All models are now evaluated using the same `_validate()` function as the training checkpoints.

4. We added analysis of schedule design, including the motivation for the linear form and a comparison with nonlinear variants.

In summary, PREPO moves beyond hand-crafted difficulty conventions and random selection in RLVR pipelines by using a model-induced approach of a prompt-perplexity schedule and rollout-entropy weighting. The revised manuscript provides additional empirical evidence and shows that reorganizing the same data can improve results without increased rollout cost. Together with the preliminary analysis, we clarify how properties of the training distribution can support more efficient RLVR training. As such, we view our work as a promising direction for online data selection when scaling RLVR training to larger datasets. We hope this motivates further study of how other data properties, such as prompt length noted by reviewer t6D8, influence RLVR training.

---

### Note · Authors · 2026-01-05

I have read and agree with the venue's withdrawal policy on behalf of myself and my co-authors.